# ASPP2 maintains the integrity of mechanically stressed pseudostratified epithelia during morphogenesis

Christophe Royer [1✉], Elizabeth Sandham[1], Elizabeth Slee[2], Falk Schneider [3,4], Christoffer B. Lagerholm[5], Jonathan Godwin[1,6], Nisha Veits [1], Holly Hathrell[1], Felix Zhou[2], Karolis Leonavicius[1,11], Jemma Garratt [1,7], Tanaya Narendra [1,7], Anna Vincent[8], Celine Jones[7], Tim Child[7,8], Kevin Coward[7], Chris Graham[7], Marco Fritzsche [9,10], Xin Lu [2] & Shankar Srinivas [1✉]

During development, pseudostratified epithelia undergo large scale morphogenetic events associated with increased mechanical stress. Using a variety of genetic and imaging approaches, we uncover that in the mouse E6.5 epiblast, where apical tension is highest, ASPP2 safeguards tissue integrity. It achieves this by preventing the most apical daughter cells from delaminating apically following division events. In this context, ASPP2 maintains the integrity and organisation of the filamentous actin cytoskeleton at apical junctions. ASPP2 is also essential during gastrulation in the primitive streak, in somites and in the head fold region, suggesting that it is required across a wide range of pseudostratified epithelia during morphogenetic events that are accompanied by intense tissue remodelling. Finally, our study also suggests that the interaction between ASPP2 and PP1 is essential to the tumour suppressor function of ASPP2, which may be particularly relevant in the context of tissues that are subject to increased mechanical stress.

[1] Department of Physiology, Anatomy and Genetics, University of Oxford, Oxford OX1 3QX, UK. [2] Ludwig Institute for Cancer Research, Nuffield Department of Medicine, University of Oxford, Oxford OX3 7DQ, UK. [3] MRC Human Immunology Unit, Weatherall Institute of Molecular Medicine, University of Oxford, Oxford, UK. [4] Translational Imaging Center, University of Southern California, Los Angeles, CA 90089, USA. [5] Wolfson Imaging Centre Oxford, MRC Weatherall Institute of Molecular Medicine, John Radcliffe Hospital, University of Oxford, Oxford, UK. [6] Department of Biochemistry, University of Oxford, South Parks Road, Oxford OX1 3QU, UK. [7] Nuffield Department of Women's and Reproductive Health, University of Oxford, Level 3, Women's Centre, John Radcliffe Hospital, Headington, Oxford OX3 9DU, UK. [8] Oxford Fertility, Institute of Reproductive Sciences, Oxford Business Park North, Oxford OX4 2HW, UK. [9] Kennedy Institute for Rheumatology, University of Oxford, Oxford OX3 7LF, UK. [10] Rosalind Franklin Institute, Didcot OX11 0QS, UK. [11]Present address: Institute of Biotechnology, Vilnius University, Vilnius, Lithuania. ✉email: christophe.royer@dpag.ox.ac.uk; shankar.srinivas@dpag.ox.ac.uk

Pseudostratified epithelia are common building blocks and organ precursors throughout embryonic development in a wide array of organisms[1]. As in other epithelia, their cells establish and maintain apical-basal polarity. However, their high nuclear density, high proliferation rate and nuclei movement during interkinetic nuclear migration (IKNM) make them unique. As IKNM proceeds, mitotic cells round-up at the apical surface of the epithelium before dividing. During this process, mitotic cells generate sufficient mechanical force to locally distort the shape of the epithelium[2] or accelerate invagination[3]. During development, pseudostratified epithelia are also subject to large-scale morphogenetic events that dramatically affect their shape and structural organisation. This is particularly true during gastrulation in the mouse embryo when cells apically constrict in the primitive streak as they push their cell body basally to eventually delaminate into the underlying mesoderm cell layer[4–7], or when the ectoderm is reshaped to form the head folds. Such morphogenesis is inextricably linked to mechanical stress (both compressive and tensile), the forces applied to the unit area to produce a change in the shape, volume or length of a cell, tissue or structure[8]. The combined mechanical stress due to IKNM and morphogenetic events poses an incredible challenge for pseudostratified epithelia to maintain tissue integrity during development. However, the molecular mechanisms that allow them to cope with increased mechanical stress are poorly defined.

During these morphogenetic events, epithelial cells continually rely on apical constrictions involving specific filamentous actin (F-actin) cytoskeleton organisation and actomyosin contractility to modify tissue shape and structural organisation[9,10]. As cells apically constrict, the coupling of apical junctions to the actomyosin network is essential in transmitting mechanical forces across tissues[11]. Reciprocally, as apical constrictions reshape the apical domain of epithelial cells, apical junctions must be able to withstand the forces generated to maintain tissue integrity. Apical-basal polarity components, such as Par3, are vital for apical constrictions[12] and the integrity of apical junctions[13]. However, it remains unknown how components of the apical-basal polarity machinery maintain tissue integrity in conditions of increased mechanical stress as morphogenetic events occur.

ASPP2 is a Par3 interactor and component of the apical junctions[14,15]. It belongs to the ASPP family of proteins, also comprising ASPP1 and iASPP, that is characterised by the common ability to interact with protein phosphatase 1 (PP1)[16,17] and a conserved C-terminal region containing ankyrin repeats, an SH3 domain and a proline-rich region. All three proteins have been described to regulate the apoptotic function of p53 in vitro[18,19]. However, ASPP2 can uniquely interact with a group of proteins that localises at the apical junctions[20] and evidence so far suggests that it exerts distinct in vivo functions from ASPP1 and iASPP[15,21–26]. ASPP2 has been shown to be able to dephosphorylate the Hippo pathway effectors YAP and TAZ via the recruitment of PP1[27,28]. In Drosophila, dASPP regulates the remodelling of adherens junctions[29–31] and its interaction with PP1 has recently been shown to be required in vivo in the context of eye patterning and wing development[32]. In mammals, ASPP2 regulates the apical-basal polarity of radial glial cells during central nervous system development[33]. Despite these findings, the in vivo function of the interaction between ASPP2 and PP1 remains unknown in mammals.

Here, using a variety of genetic and imaging approaches, we demonstrate that during morphogenetic events crucial for the normal development of the early post-implantation embryo, ASPP2 maintains epithelial structural integrity in pseudostratified epithelia under increased mechanical stress. ASPP2 is required for the maintenance of proamniotic cavity and primitive streak architecture, somite structure and head fold formation. In the proamniotic cavity, ASPP2 maintains epithelial architecture by preventing apical daughter cells from escaping the epiblast. Mechanistically, we further demonstrate that this requires the PP1-binding site of ASPP2 and it is achieved through ASPP2's key role in maintaining F-actin cytoskeleton organisation at the apical junctions. Our results show that ASPP2 is an essential component of a system that maintains tissue integrity under conditions of increased mechanical stress in a broad range of tissues.

## Results

**ASPP2 is not required for TE development.** ASPP2 can regulate both apical-basal cell polarity and the phosphorylation status of YAP/TAZ through its interaction with Par3[14,15] and PP1[27,28] respectively. Both cell polarity and the phosphorylation of YAP and TAZ are crucial to TE development[34–38]. We thus started with the hypothesis that ASPP2 may be important for outside cell polarisation and TE fate determination during preimplantation development. In support of this, we found that ASPP2 could start to be detected as early as E2.5 at cell-cell junctions (Supplementary Fig. 1a). As seen in other examples of polarised epithelia[14,15,28,39], ASPP2 was strongly localised to the apical junction in the trophectoderm from the 32-cell stage onwards (Fig. 1a). This localisation pattern was similar in human blastocysts, suggesting that ASPP2 behaves in a similar way across mammals (Supplementary Fig. 1b).

A previous study revealed that ASPP2 may play a role during early embryogenesis, as ASPP2 mutant embryos in which exons 10–17 of Trp53bp2 (ASPP2) were deleted could not be recovered at E6.5[40]. Nevertheless, the phenotype of these embryos was not described, and earlier stages were not examined. To investigate the role of ASPP2 during preimplantation development, we generated ASPP2-null embryos in which exon 4 was deleted ($ASPP2^{\Delta E4/\Delta E4}$) resulting in a frameshift and early stop codons. To be able to distinguish between phenotypes relating to ASPP2's PP1 regulatory function and other functions, we also generated embryos homozygous for a mutant form of ASPP2 that was specifically unable to associate with PP1 ($ASPP2^{RAKA/RAKA}$)[17]. $ASPP2^{\Delta E4/\Delta E4}$ and $ASPP2^{RAKA/RAKA}$ blastocysts appeared to be morphologically normal with properly formed blastocyst cavities (Fig. 1b and Supplementary Fig. 1c). YAP was clearly nuclear in the TE and cytoplasmic in the ICM suggesting that the ICM and TE lineage were properly allocated (Fig. 1b). The polarity protein Par3 (Fig. 1b) and F-actin cables (Supplementary Fig. 1c) were strongly localised at the apical junctions, suggesting that polarity and overall cell architecture were normal. It was sometimes possible to see some residual ASPP2 protein at the apical junction in early $ASPP2^{\Delta E4/\Delta E4}$ blastocysts, potentially due to residual maternal ASPP2 expression. To eliminate the possibility that perdurance of maternally encoded ASPP2 compensated for the zygotic mutations, we microinjected a one-cell embryo with siRNA against ASPP2 and cultured them to the blastocyst stage. Control and ASPP2-depleted embryos were morphologically indistinguishable. The localisation of YAP was similar between control and ASPP2-depleted embryos (Fig. 1c). YAP phosphorylation at Serine 127 was stronger in the cytoplasm of ICM cells in comparison to the TE in both controls and ASPP2-depleted embryos (Supplementary Fig. 1d). Taken together, these results show that neither ASPP2's polarity function nor its PP1 regulatory function is required during preimplantation development.

**ASPP2 is required for proamniotic cavity architecture.** Since it has previously been shown that deletion of ASPP2 may be embryonic lethal around E6.5[40], we next investigated whether

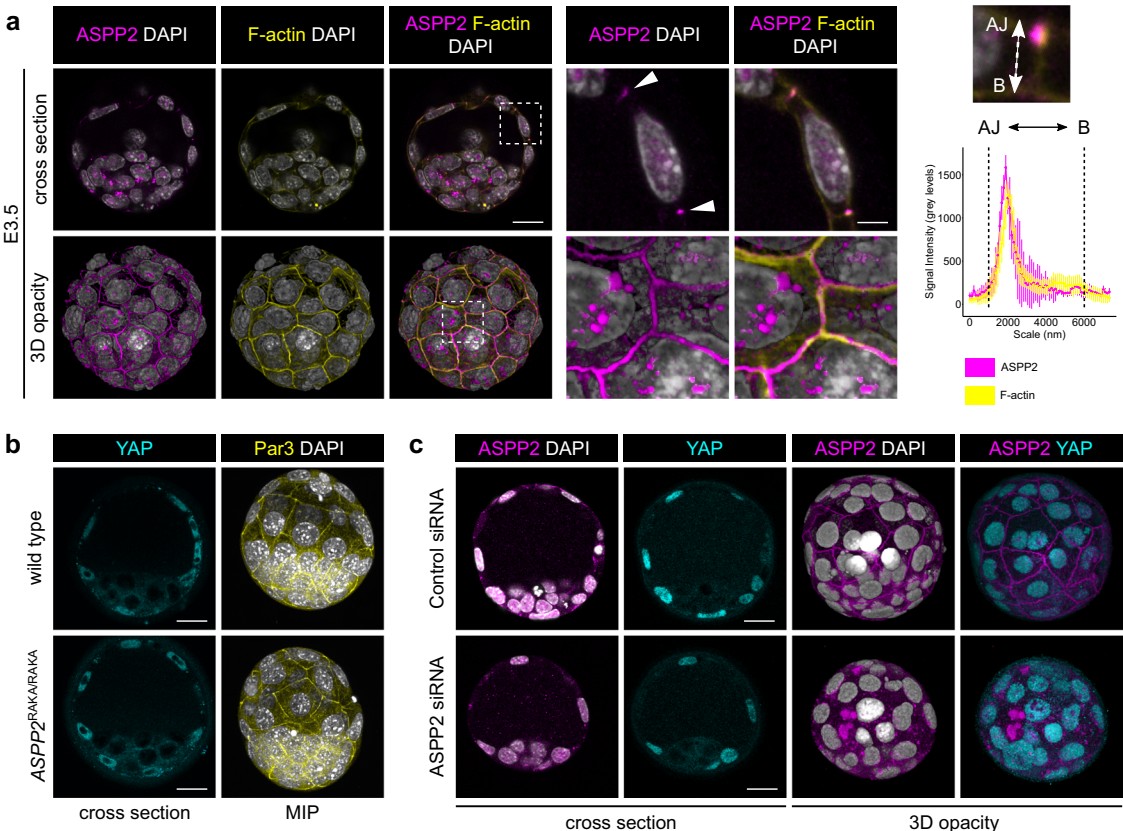

**Fig. 1 The ASPP2/PP1 complex is not required during preimplantation development. a** ASPP2 was detected by indirect immunofluorescence in E3.5 embryos to analyse its localisation pattern. A cross-section through the equatorial plane of a representative embryo is shown (top row), as well as a 3D opacity rendering of the same embryo (bottom row). The F-actin cytoskeleton and nuclei were visualised using Phalloidin and DAPI, respectively. A magnified image of the dashed area is shown on the right. Note how ASPP2 colocalises with F-actin at the apical junctions in cells of the trophectoderm (white arrowheads). The juxtaposed graph shows ASPP2 and F-actin signal intensity along the apical-basal axis of five cell-cell junctions in the TE. Error bars represent ±SD. AJ apical junction, B base of the trophectoderm. Scale bars: 20 and 5 μm (for the magnification). **b** The localisation pattern of YAP and Par3 was analysed in wild type and $ASPP2^{RAKA/RAKA}$ embryos by indirect immunofluorescence. A cross-section of representative embryos through the equatorial plane shows the localisation of YAP in the nuclei of the trophectoderm in both wild type and $ASPP2^{RAKA/RAKA}$ embryos. Maximum intensity projections of these embryos show the localisation of Par3 at the level of apical junctions in the trophectoderm (representative images from six wild type and eight $ASPP2^{RAKA/RAKA}$ embryos). Scale bar: 20 μm. **c** ASPP2 knockdown in E3.5 embryos using siRNA against ASPP2 mRNA. ASPP2 knockdown was confirmed by indirect immunofluorescence. Note how signal at the apical junctions is specific to ASPP2 and how YAP is normally localised to the nuclei of TE cells in ASPP2-depleted embryos. Representative images from $n = 19$ control siRNA-injected embryos and $n = 24$ ASPP2 siRNA-injected embryos across two independent experiments. Scale bar: 20 μm. Source data are provided as a Source Data file.

ASPP2 was required at early post-implantation stages. To address this, we generated $ASPP2^{\Delta E4/\Delta E4}$ embryos at different stages and examined the localisation of Par6 and F-actin to assess apical-basal polarity and overall tissue organisation, respectively. At E5.5, $ASPP2^{\Delta E4/\Delta E4}$ embryos did not exhibit obvious morphological defects in comparison to wild type and heterozygous littermates. Polarised Par6 could be detected at the apical membrane in the visceral endoderm (VE) and the epiblast, suggesting that both cell layers were properly polarised (Supplementary Fig. 2a). In contrast, E6.5 $ASPP2^{\Delta E4/\Delta E4}$ embryos exhibited strong morphological defects in comparison to wild type and heterozygous littermates. The proamniotic cavity was either absent (six embryos out of ten mutants) or was greatly reduced in size at E6.5 (four embryos out of ten mutants) (Fig. 2a) and always absent at E7.5 (Supplementary Fig. 2b). In embryos lacking a proamniotic cavity, the epiblast was disorganised, and instead of being a pseudostratified epithelium, appeared multi-layered. This seemed to be the result of an ectopic accumulation of cells from the epiblast in place of the proamniotic cavity. The ectopic cells in the centre of the embryo exhibited a complete lack of polarised Par6 and in embryos with reduced cavity size, the

accumulated cells showed reduced apical Par6 (Fig. 2b). This suggested that the ectopic accumulation of cells where the proamniotic cavity ought to be was accompanied by a progressive loss of cell polarity in the epiblast. F-actin localisation was also profoundly abnormal in these cells (Fig. 2b). In wild type embryos, F-actin was enriched at the apical junctions of epiblast cells, whereas in $ASPP2^{\Delta E4/\Delta E4}$ embryos it was distributed more uniformly across the apical surface (Fig. 2c). This suggests that ASPP2 is required for organising the F-actin cytoskeleton at the apical junctions.

Because signals from the basement membrane are believed to be essential for proamniotic cavity formation[41], we examined the localisation of laminin and found it unaltered in $ASPP2^{\Delta E4/\Delta E4}$ embryos. This suggested that the loss of cell polarity and ectopic accumulation of cells was not accompanied by, or due to, breakage in the basement membrane (Fig. 2d, e). Finally, when we examined the basolateral membrane marker SCRIB, we observed that its basolateral localisation was unaffected in $ASPP2^{\Delta E4/\Delta E4}$ embryos. However, SCRIB was also strongly expressed at the apical junctions in the epiblast of wild type embryos. This particular localisation pattern was intermittently disrupted in

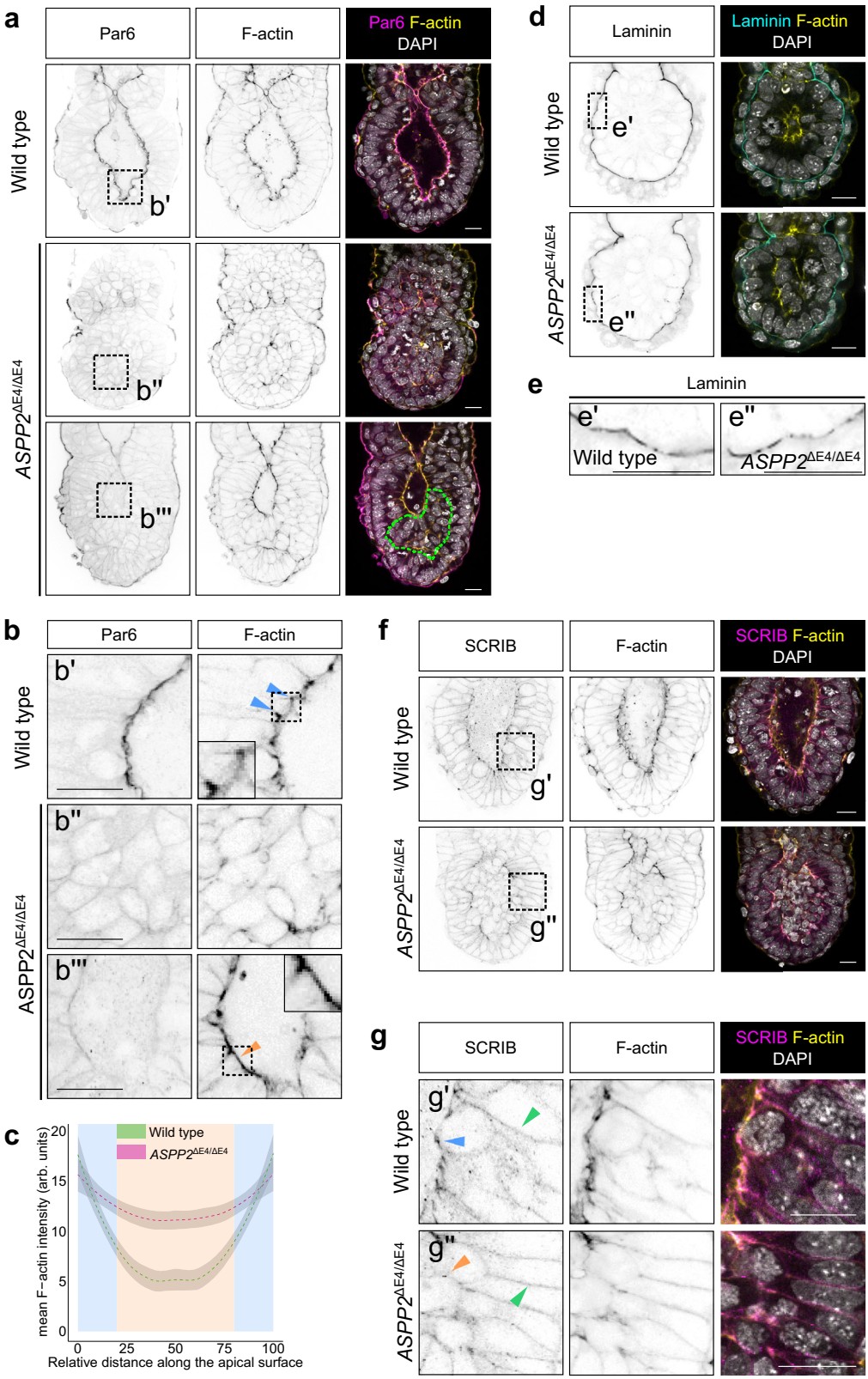

$ASPP2^{\Delta E4/\Delta E4}$ embryos, specifically at the interface between cells of the epiblast and cells ectopically accumulating in the proamniotic cavity (Fig. 2f, g). Together, these results suggest that the apparent loss of apical cell polarity seen in $ASPP2^{\Delta E4/\Delta E4}$ embryos originates from defects specific to the apical junctions rather than at the level of the basolateral or basement membranes.

The VE monolayer epithelium that forms the outside layer of the egg cylinder also normally expresses ASPP2 (see below and Supplementary Fig. 6a, b). This tissue appeared normal in $ASPP2^{\Delta E4/\Delta E4}$ mutants, expressed the pan-VE marker GATA6 indicating that it was correctly specified, and exhibited apical Par6 (Fig. 2a and Supplementary Fig. 2c), suggesting that its epithelial

**Fig. 2 Absence of ASPP2 expression leads to structural defects in the epiblast. a** Immunofluorescence of wild type and *ASPP2*$^{\Delta E4/\Delta E4}$ E6.5 embryos using an anti-Par6 antibody. The phenotypic variability of *ASPP2*$^{\Delta E4/\Delta E4}$ embryos is illustrated, with embryos either lacking cavities (middle row, five out of nine embryos) or exhibiting smaller cavities (bottom row, four out of nine embryos). The green dashed line highlights the ectopic accumulation of cells in the epiblast of *ASPP2*$^{\Delta E4/\Delta E4}$ embryos. **b** Magnification of the corresponding regions shown in panel **a**. Blue arrowheads highlight the enrichment of F-actin at the apical junctions in the epiblast. Note how F-actin is not enriched at the apical junctions but is instead more homogenously distributed across the apical surface of epiblast cells in *ASPP2*$^{\Delta E4/\Delta E4}$ embryos (orange arrowhead). The insets within images are 2x magnifications of the corresponding dashed areas. **c** Quantification of F-actin signal intensity along the apical surface of epiblast cells of wild type (*n* = 3 embryos, five measurements per embryo) and *ASPP2*$^{\Delta E4/\Delta E4}$ embryos (*n* = 3 embryos, five measurements per embryo). Measurements were made on cross-sections along the apical domain of individual epiblast cells from apical junction to an apical junction (represented with a blue background in the graph). The 95% confidence interval is represented by the grey area. See material and methods for details. **d** Immunofluorescence of wild type (representative images from eight embryos) and *ASPP2*$^{\Delta E4/\Delta E4}$ E5.5 embryos (representative images from five embryos) using an anti-Laminin antibody. **e** Magnification of the corresponding dashed areas in panel **d**. **f** Immunofluorescence of wild type (representative images from seven embryos) and *ASPP2*$^{\Delta E4/\Delta E4}$ (representative images from two embryos) E6.5 embryos using an anti-SCRIB antibody. **g** Magnification of the corresponding dashed areas in panel **f**. Green arrowheads highlight basolateral SCRIB. Note the enrichment of SCRIB at the apical junctions in the epiblast of wild type embryos (blue arrowhead) and its absence in the corresponding localisation in *ASPP2*$^{\Delta E4/\Delta E4}$ embryos (orange arrowhead). Nuclei and the F-actin cytoskeleton were visualised with DAPI and Phalloidin, respectively. Scale bars: 20 μm. Source data are provided as a Source Data file.

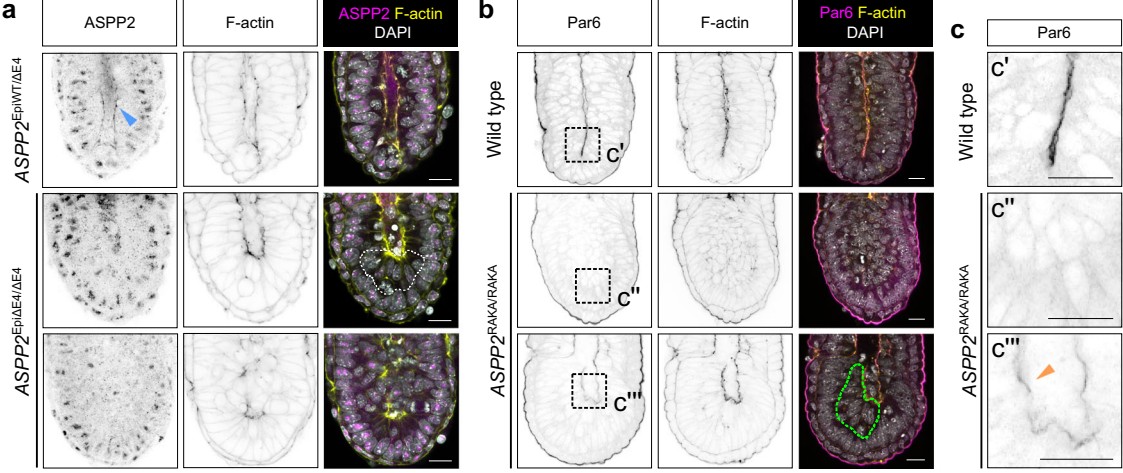

**Fig. 3 *ASPP2*$^{Epi\Delta E4/\Delta E4}$ and *ASPP2*$^{RAKA/RAKA}$ embryos phenocopy *ASPP2*$^{\Delta E4/\Delta E4}$ embryos. a** The expression of ASPP2 was conditionally ablated in the epiblast to test for its epiblast-specific requirement (*ASPP2*$^{Epi\Delta E4/\Delta E4}$ embryos). The ASPP2 expression pattern was analysed by indirect immunofluorescence in *ASPP2*$^{EpiWT/\Delta E4}$ (representative images from four embryos) and *ASPP2*$^{Epi\Delta E4/\Delta E4}$ (representative images from four embryos) embryos. ASPP2 proteins were completely absent at the apical junction of epiblast cells in *ASPP2*$^{Epi\Delta E4/\Delta E4}$ embryos. Note that the ASPP2 antibody results in a nonspecific nuclear signal (also seen in Fig. 1c when depleting ASPP2 by siRNA). The dashed area highlights the ectopic accumulation of cells in the epiblast. **b** Immunofluorescence of wild type (representative images from seven embryos) and *ASPP2*$^{RAKA/RAKA}$ (representative images from three embryos) E6.5 embryos using an anti-Par6 antibody. The green dashed line highlights the ectopic accumulation of cells in the epiblast of *ASPP2*$^{RAKA/RAKA}$ embryos. **c** Magnification of the corresponding dashed regions in **b**. Note the reduced amount of Par6 along the apical domain of epiblast cells in *ASPP2*$^{RAKA/RAKA}$ embryos (orange arrowhead). Nuclei and the F-actin cytoskeleton were visualised with DAPI and Phalloidin, respectively. Scale bars: 20 μm.

architecture was maintained. The localisation of F-actin and Par6 in the extraembryonic ectoderm (EXE) was also normal at E6.5 (Supplementary Fig. 2d). Together this suggests that at E6.5, ASPP2 is required specifically in the epiblast.

To verify that an epiblast-specific requirement for ASPP2 led to the proamniotic cavity defects in *ASPP2*$^{\Delta E4/\Delta E4}$ embryos, we conditionally ablated ASPP2 expression in just the epiblast (*ASPP2*$^{Epi\Delta E4/\Delta E4}$ embryos) (Fig. 3a). These embryos phenocopied *ASPP2*$^{\Delta E4/\Delta E4}$ embryos, demonstrating that the observed phenotype can be traced to a requirement for ASPP2 in only the epiblast. To test if the accumulation of cells in the proamniotic cavity is due to a loss of adhesion between epiblast cells, we visualised E-cadherin localisation. As in controls, in mutants, E-cadherin was distributed basolaterally as normal amongst cells of the epiblast and was intermittently disrupted only at the interface between cells of the epiblast and cells abnormally accumulating in the proamniotic cavity, again pointing to a defect specific to the apical junctions (Supplementary Fig. 2e).

We also tested whether a differential rate of proliferation or cell death could explain the proamniotic cavity defect in these embryos. We found no significant difference in the proportion of Phospho-Histone H3 (PHH3) positive cells in the epiblast of control and *ASPP2*$^{Epi\Delta E4/\Delta E4}$ embryos (Supplementary Fig. 2f). Very few cells in the epiblast of either control or *ASPP2*$^{Epi\Delta E4/\Delta E4}$ embryos exhibited cleaved Caspase-3 at E6.5 (Supplementary Fig. 2g), arguing against cell death plays a role in the phenotype. We noted that at E7.5, when the phenotype was severe, cleaved Caspase-3 positive cells that could be detected in mutants were located amongst the cells abnormally accumulating in the centre of embryos, presumably due to the high cell density (Supplementary Fig. 2g). Despite this apparent loss of epiblast architecture at E7.5, mesoderm and definitive endoderm specification could still occur in ASPP2 mutant embryos, as shown by the localised expression of T in the posterior epiblast (Supplementary Fig. 2b) and of SOX17 throughout the outside cell layer (Supplementary Fig. 2g).

To test whether this requirement for ASPP2 in proamniotic cavity morphogenesis is rooted in its ability to recruit and regulate PP1, we analysed $ASPP2^{RAKA/RAKA}$ embryos at E6.5. We found that $ASPP2^{RAKA/RAKA}$ embryos, similarly to $ASPP2^{\Delta E4/\Delta E4}$ embryos, exhibit either reduced proamniotic cavity size or no cavity at all. This was again accompanied by a reduced apical Par6 in the epiblast when the proamniotic cavity was of reduced size and the absence of apical Par6 when no cavity was present (Fig. 3b, c). The basolateral localisation of SCRIB once again was not affected, whereas its localisation at the apical junctions was severely disrupted in $ASPP2^{RAKA/RAKA}$ embryos (Supplementary Fig. 2i, j). This shows that at E6.5, $ASPP2^{RAKA/RAKA}$ mutant embryos have an identical phenotype to $ASPP2^{\Delta E4/\Delta E4}$ mutant embryos, demonstrating the key role of the PP1-binding site of ASPP2 in regulating epiblast and proamniotic cavity architecture.

**ASPP2 controls epiblast apical daughter cell reincorporation.** Our results so far show that ASPP2 is essential for the architecture of the epiblast and the maintenance of the proamniotic cavity. When ASPP2's function is impaired, apolar cells accumulate ectopically in place of the proamniotic cavity. However, it remained unclear how this occurred and what biological process ASPP2 actually controls in the epiblast. Amongst possible explanations is that the phenotype was the consequence of epiblast cells delaminating apically into the proamniotic cavity because of a drastic shift in the proportion of orthogonal cells divisions, a breakdown of the apical junction domain, a failure of daughter cells reincorporating basally following cell divisions or a combination of these. To answer this question, we generated $ASPP2^{\Delta E4/\Delta E4}$ embryos with fluorescently labelled membranes, which enabled us to follow the movement of epiblast cells in these embryos by time-lapse confocal microscopy (Fig. 4a and Supplementary Movie 1). In wild type and heterozygous embryos, we could observe the movement of cell bodies along the apical-basal axis during interkinetic nuclear migration (INM), with mitotic cells rounding up at the apical surface of the epiblast before dividing as previously described[42].

We first analysed the orientation of cell divisions but could not detect differences in overall cell division angle in $ASPP2^{\Delta E4/\Delta E4}$ embryos in comparison to controls (Supplementary Fig. 3a), even when division events were binned into categories as 'orthogonal', 'parallel' or 'oblique'. To determine if there was a defect in interkinetic nuclear migration, we also analysed the distance at which cell divisions occurred from the basement membrane of the epiblast. Again, there was no notable difference between $ASPP2^{\Delta E4/\Delta E4}$ and wild type embryos, suggesting that even in the absence of apical-basal polarity and the proamniotic cavity, cells of the epiblast were able to proceed with INM (Supplementary Fig. 3b).

We next investigated the behaviour of daughter cells after cytokinesis. In wild type embryos, following cell divisions, the cell body of both daughters moved basally so that they came to span the entire height of the epithelium along the apical-basal extent of the epiblast. In contrast, in $ASPP2^{\Delta E4/\Delta E4}$ embryos, dividing cells moved towards the embryonic centre as normal, but upon division, daughter cells delaminated apically towards the centre of mass of the embryonic region (Fig. 4a). This suggested that ASPP2 may be specifically required for the retention of daughter cells within the epiblast. To further characterise this failure of dividing cells to reintegrate into the epiblast epithelium, we quantified the movement of daughter cells from the initial point of cell division (Fig. 4b–d). We found that in wild type and heterozygous embryos, this movement was almost always basal for both daughters (51/56, 91.1%). In contrast, in mutant embryos, for half the daughter cells (33/66, 50%) the movement

was apical (Fig. 4c, d). We found that in the majority of cases (29/33, 87.9%), it was the daughter that was relatively more apically positioned with respect to its sister that abnormally moved apically following cell divisions (Fig. 4d). This suggests that ASPP2 is involved in a mechanism specifically required for apical daughter cell reintegration into the pseudostratified epiblast following cell division, which is crucial in maintaining the architecture of the epiblast and proamniotic cavity.

**ASPP2 is required in regions of high mechanical stress.** Our results all point to an important role for ASPP2 in regulating tissue architecture, possibly via the regulation of F-actin organisation at the apical junction. However, this was difficult to study in $ASPP2^{\Delta E4/\Delta E4}$ and $ASPP2^{RAKA/RAKA}$ embryos on a C57BL/6 background because of the relative severity of the defect. We therefore bred $ASPP2^{RAKA/RAKA}$ mutation into a BALB/c background, to take advantage of the fact that ASPP2 phenotypes are often not as dramatic in this background[15]. Consistent with this, BALB/c $ASPP2^{RAKA/RAKA}$ homozygous embryos completely bypassed the phenotype at E6.5 observed in C57BL/6 $ASPP2^{RAKA/RAKA}$ embryos. Instead, the phenotype of these embryos was milder, and they were only grossly different from wild type and heterozygous embryos 1 day later, at E7.5. $ASPP2^{RAKA/RAKA}$ embryos exhibited two distinct phenotypes. The majority (34/41, 82.9%), that we termed type I embryos, exhibited a strong accumulation of cells in their posterior, suggestive of a defect in the primitive streak (Fig. 5a and Supplementary Fig. 4a). A minority (7/41, 17.1%), that we termed type II embryos, were developmentally delayed but did not exhibit any structural defects. Importantly, none of these defects were a result of ASPP2RAKA mutant proteins being unstable or mislocalised as they could be observed at the apical junctions in embryos at similar levels to wild type ASPP2 (Supplementary Fig. 4b).

Given that $ASPP2^{\Delta E4/\Delta E4}$ and $ASPP2^{RAKA/RAKA}$ in a Bl/6 background showed striking abnormalities in the localisation of F-actin, we examined the localisation of F-actin in the posterior of type I $ASPP2^{RAKA/RAKA}$ embryos at E7.5 (Fig. 5b and Supplementary Fig. 4c). In wild type embryos, we were able to clearly identify cells apically constricting and pushing their cell body basally towards the nascent mesodermal cell layer. This is characteristic of cells in the primitive streak in the process of delaminating basally[4]. It was also evident that F-actin was enriched at the apical junctions in these cells (Fig. 5c). In contrast, $ASPP2^{RAKA/RAKA}$ embryos exhibited a clear ectopic accumulation of cells apical to the primitive streak as visualised by T expression (Fig. 5b). In these cells, F-actin was abnormally uniformly disturbed along the apical surface, with no clear apical-junction enrichment. This was specific to cells in the posterior of the epiblast where the primitive streak forms, as F-actin organisation was undisturbed in the lateral regions of the epiblast (Fig. 5c). To investigate the localisation of F-actin in more detail, we performed Airyscan super-resolution imaging of these embryos (Fig. 5d–f). This revealed that the mesh-like structure normally formed by F-actin at the apical junctions in cells of the epiblast was severely disrupted in the posterior of $ASPP2^{RAKA/RAKA}$ embryos. Instead, F-actin appeared to form spike-like structures at the surface of cells in this region (Supplementary Movie 2 and Fig. 5e, f). This profound disruption of F-actin localisation indicates that the PP1 regulatory function of ASPP2 is required for F-actin organisation in the cells of the primitive streak.

The epiblast-specific requirement for ASPP2 despite its broad expression (Fig. 1a and see below) and the localisation of the phenotype primarily to the posterior region of the epiblast in the BALB/c background suggest that specific epithelia or epithelial

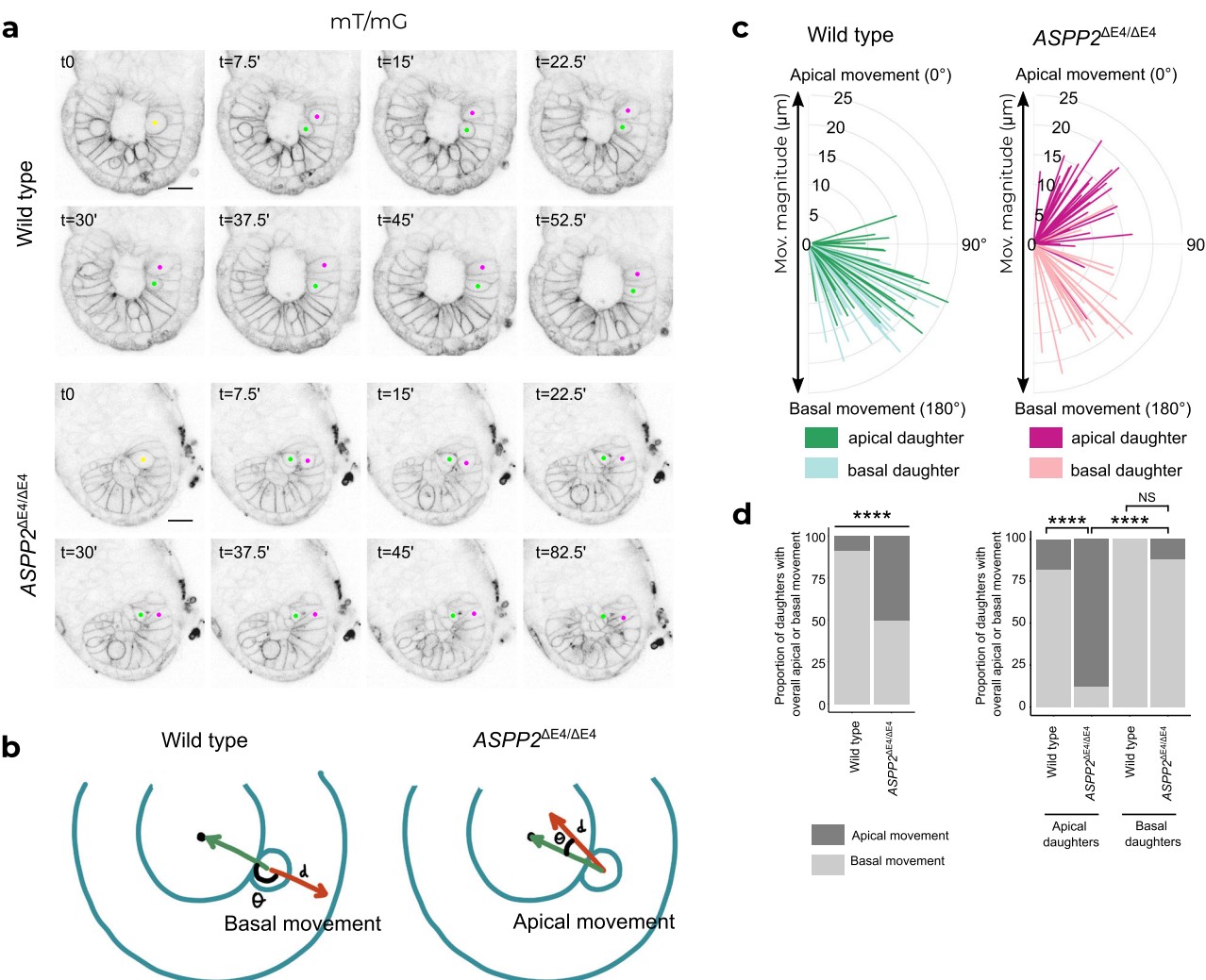

**Fig. 4 ASPP2 is required for apical daughter cell reincorporation into the epiblast following cell division events. a** Time-lapse imaging of wild type and $ASPP2^{\Delta E4/\Delta E4}$ embryos. mT/mG-labelled cell membranes were used to manually track cell movement. Yellow dots highlight mother cells at the apical surface of the epiblast immediately prior to a cell division event. Green and magenta dots identify the resulting daughter cells. Note how both daughters reintegrate the epiblast in the wild type whereas one of the two daughters fails to do so in the absence of ASPP2 even after a prolonged period of time ($t = 82.5'$). **b** Diagram illustrating the method used to quantify daughter cell movement following cell divisions. Daughter cell movement was characterised by both the distance travelled ($d$) and the direction of travel ($\theta$) expressed as the angle between the reference vector (the green vector starting from the initial position of the mother cell prior to the division event to the centre of the embryonic region) and the vector characterising absolute daughter cell movement (the red vector starting from the initial position of the mother cell prior to the division event to the final position of the daughter cell). The left panel illustrates the case of a daughter moving basally to reincorporate the epiblast and the right panel describes abnormal daughter cell movement towards the centre of the embryonic region such as seen in $ASPP2^{\Delta E4/\Delta E4}$ embryos. **c** Graph quantifying cell movement in wild type ($n = 3$ embryos, 56 cells) and $ASPP2^{\Delta E4/\Delta E4}$ embryos ($n = 3$ embryos, 66 cells). For a given pair of daughter cells, each daughter was defined as 'apical' or 'basal' depending on their respective position relative to the centre of the embryonic region immediately after a cell division event. **d** Proportion of daughter cells with an overall apical or basal movement in wild type ($n = 3$ embryos, 56 cells) and $ASPP2^{\Delta E4/\Delta E4}$ embryos ($n = 3$ embryos, 66 cells). Left panel: Quantification of the proportion of daughter cells with an overall apical ($\theta$ from 0° to 90°) or basal movement ($\theta$ from 90° to 180°) in wild type and $ASPP2^{\Delta E4/\Delta E4}$ embryos. Right panel: quantification of the proportion of apical and basal daughters with an overall apical ($\theta$ from 0° to 90°) or basal movement ($\theta$ from 90° to 180°) in wild type and $ASPP2^{\Delta E4/\Delta E4}$ embryos. ****$p < 0.0001$, NS non-significant (two-sided Fisher's exact test of independence. The Bonferroni method was used to adjust $p$ values for multiple comparisons. $P$ values from left to right: $p = 6.16e-07$, $p = 1.62e-08$, $p = 2.83e-09$, $p = 7.08e-01$). Source data are provided as a Source Data file.

regions are more sensitive to ASPP2 deficiency than others. We hypothesised therefore that ASPP2 may be important particularly within epithelia subject to increased mechanical stress at the apical junction, for example, during the apical curving required to form a cavity. This hypothesis predicts that the apical domain of the epiblast during proamniotic cavity formation would be subject to elevated stress.

To evaluate this prediction, we quantified the relative apical mechanical stress[8] of different tissues in E6.5 embryos, using the FLIPPER-TR membrane tension-sensitive probe, in conjunction with fluorescence lifetime imaging microscopy (FLIM)[43]. A longer lifetime of the probe corresponds to a membrane environment under higher tension. Fluorescence lifetimes were significantly higher at the apical surface of the epiblast (Mean = 4.83 ns, SD = 0.056 ns) in comparison to the apical surface of the embryonic VE (emVE) (Mean = 4.5 ns, SD = 0.089 ns) or extra-embryonic VE (exVE) (Mean = 4.65 ns, SD = 0.08 ns). This suggests that at this stage of development, apical tension is

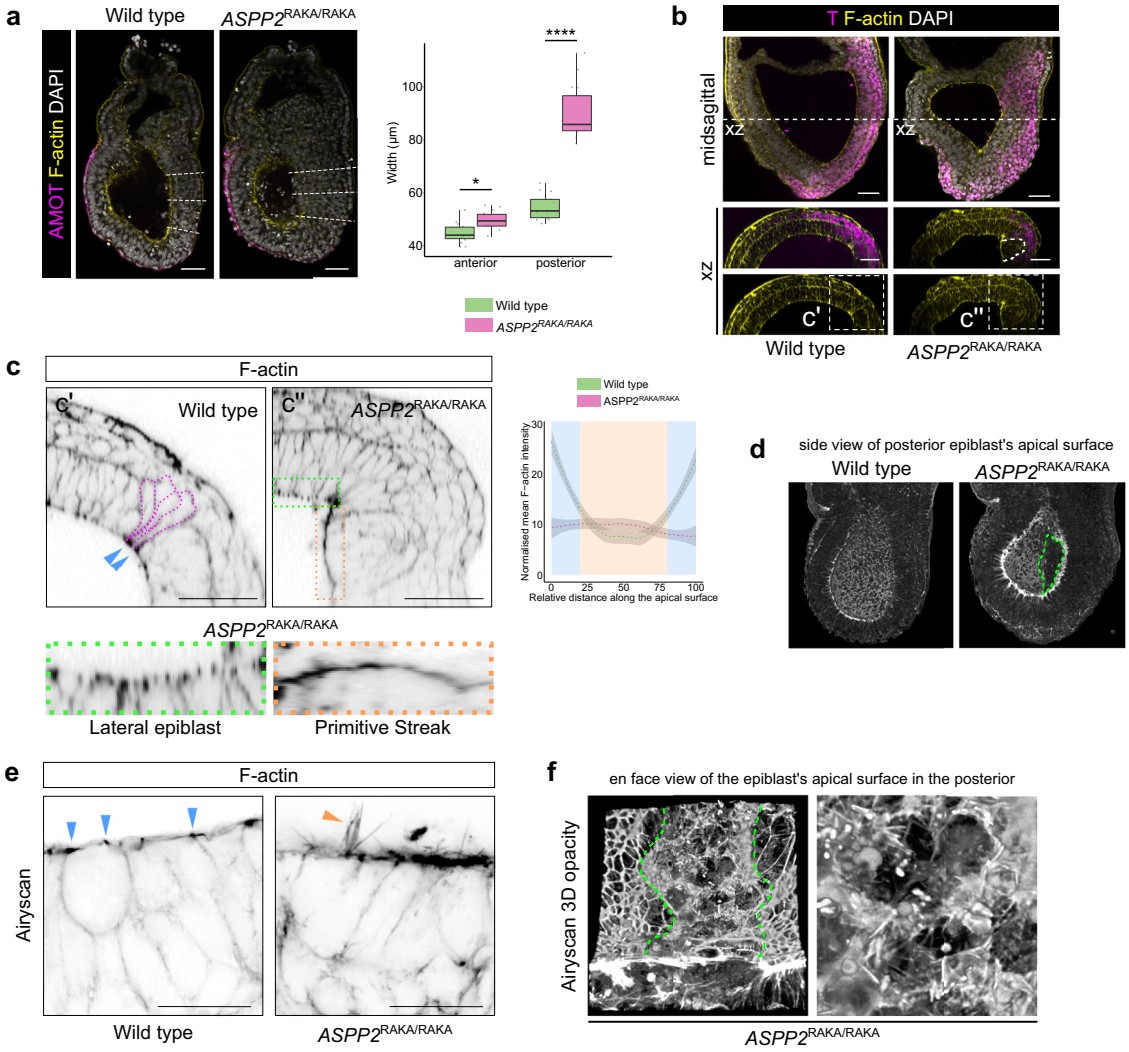

**Fig. 5 ASPP2 is required for epithelial integrity in the primitive streak. a** Posterior thickening in E7.5 *ASPP2*^RAKA/RAKA embryos in a BALB/C background. Left panel: the anteroposterior axis was defined using AMOT localisation pattern. Right panel: comparison of tissue thickness in the anterior (three measurements per embryo) and the posterior (three measurements per embryo) of wild type ($n = 5$ embryos) and *ASPP2*^RAKA/RAKA embryos ($n = 5$ embryos). For the box plots, the top and bottom lines of each box represent the 75th and 25th percentiles, respectively. The whiskers show the minima to the maxima values and the central line indicates the median. *$p < 0.05$, ****$p < 0.0001$ (nested ANOVA, *p* values from left to right: $p = 0.047$, $p = 3.95e-5$). **b** Cells accumulate in the primitive streak region of *ASPP2*^RAKA/RAKA embryos. Immunofluorescence of E7.5 wild type (representative images from 55 embryos) and *ASPP2*^RAKA/RAKA (representative images from 23 embryos) embryos using a T (Brachyury) antibody. **c** Cells ectopically accumulating in the primitive streak region are unable to apically constrict and do not have enriched F-actin at the apical junctions (area delineated by the dotted orange line) in comparison to wild type (blue arrowheads and magenta dotted lines). Green dotted ROI: the apical surface of the epiblast in the lateral region of the embryo. Lower panel: Magnified regions highlighted in green and orange, respectively. Right panel: quantification of F-actin signal intensity along the apical surface of epiblast cells in the primitive streak region of wild type ($n = 3$ embryos, five cells per embryo) and *ASPP2*^RAKA/RAKA embryos ($n = 3$ embryos, five cells per embryo). The 95% confidence interval is represented by the grey area. **d**–**f** Airyscan imaging reveals the extent of F-actin disorganisation at the surface of cells accumulating ectopically in the primitive streak region of *ASPP2*^RAKA/RAKA embryos. **d** 3D opacity rendering of embryo optical halves, enabling visualisation of the apical surface of epiblast cells in the proamniotic cavity. Note the absence of the typical F-actin mesh pattern at the apical surface of cells in the posterior of *ASPP2*^RAKA/RAKA embryos (green dotted line). **e** Cross-section through the primitive streak region, showing enriched F-actin at the apical junctions of wild type (representative image from three embryos) embryos (blue arrowheads) and the formation of F-actin spike-like structures at the contact-free surface of *ASPP2*^RAKA/RAKA (representative image from three embryos) embryos. **f** En face view of the epiblast's apical surface in the posterior of an *ASPP2*^RAKA/RAKA embryo. Green dotted lines demarcate the disorganised apical region of the posterior and the more organised lateral regions of the epiblast. Right panel: magnification of the epiblast's apical surface in the posterior of an *ASPP2*^RAKA/RAKA embryo showing F-actin forming spike-like structures. Nuclei and the F-actin cytoskeleton were visualised with DAPI and Phalloidin, respectively. Scale bars: 50 μm (**a**–**c**), 20 μm (**e**). Source data are provided as a Source Data file.

higher in the epiblast in comparison to other tissues (Fig. 6a, b and Supplementary Fig. 5). These lifetime differences are in the same order of magnitude as in aspiration pipette experiments performed on individual giant unilamellar vesicles[43] and therefore represent substantial tension differences. We also found

additional evidence that the organisation of the apical junction in the epiblast differs from that of other tissues. SHROOM2, a protein that binds F-actin, Myosin and ZO-1 at tight junctions[44], was specifically enriched at the apical junctions of the epiblast (Fig. 6c). Considering the importance of SHROOM proteins in

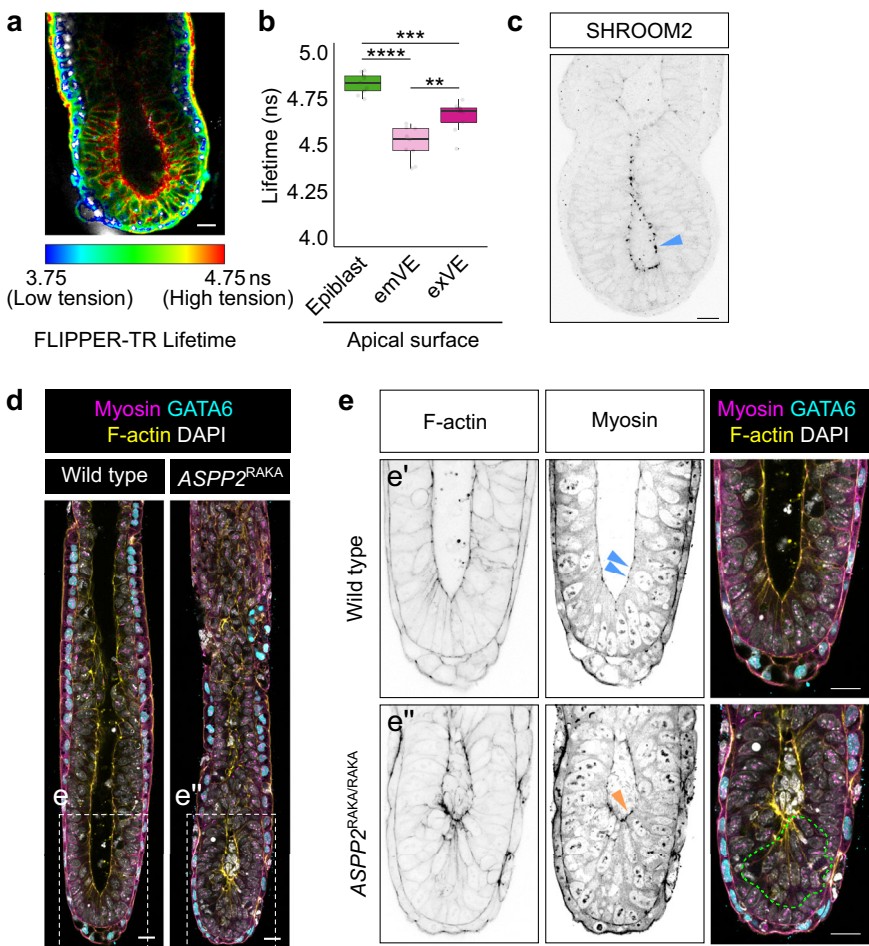

**Fig. 6 *ASPP2*<sup>RAKA/RAKA</sup> embryo are more susceptible to mechanical stress. a, b** FLIM measurements of the FLIPPER-TR tension probe in E6.5 embryos. **a** Representative FLIM image of an E6.5 embryo. Note that Lifetime smaller than 3.75 and higher than 4.75 are blue and red, respectively. **b** Mean lifetime values at the apical surface of the epiblast, exVE and emVE ($n = 9$ embryos). For the box plots, the top and bottom lines of each box represent the 75th and 25th percentiles, respectively. The whiskers show the minima to the maxima values and the central line indicates the median. ****$p < 0.0001$, ***$p < 0.001$, **$p < 0.01$ (ANOVA, followed by Tukey's test. *P* values from left to right: $p = 0$, $p = 1.52e\text{-}4$, $p = 1.41e\text{-}3$). **c** Localisation pattern of SHROOM2 in E6.5 embryos. Blue arrowhead highlights the accumulation of SHROOM2 at the apical junctions in the epiblast. **d** wild type ($n = 4$) and *ASPP2*<sup>RAKA/RAKA</sup> ($n = 2$) embryos were grown for 30' in cylindrical cavities made of biocompatible hydrogels. The localisation pattern of GATA6 and Myosin was then analysed by immunofluorescence. **e** Magnification of the embryos shown in **b**. The green dotted line highlights the ectopic accumulation of cells seen in *ASPP2*<sup>RAKA/RAKA</sup> embryos. Note how Myosin is enriched at the apical junctions of wild type epiblast cells (blue arrowheads). The orange arrowhead points to the abnormal distribution of Myosin at the apical surface of these cells. Nuclei and the F-actin cytoskeleton were visualised with DAPI and Phalloidin, respectively. Scale bars: 20 μm. Source data are provided as a Source Data file.

regulating apical actomyosin contractility[45], this suggested that the higher apical tensions observed in the epiblast may be in response to these cells apically constricting.

One prediction of the hypothesis that ASPP2 is required in places of increased mechanical stress is that subjecting Type I *ASPP2*<sup>RAKA/RAKA</sup> embryos in a BALB/c background to increased mechanical stress might induce an earlier or more severe phenotype reminiscent to that seen in the C57BL/6 background. To test this prediction, we cultured E6.5 BALB/c wild type and mutant embryos (a day before any phenotype is evident in mutants) within the confines of cylindrical cavities made of biocompatible hydrogels, in order to alter their shape[46] and subject the epiblast epithelium to higher levels of mechanical stress (Fig. 6d). Wild type embryos elongated without showing any sign of disrupted tissue integrity (four out of four embryos). Conversely, *ASPP2*<sup>RAKA/RAKA</sup> embryos showed reduced cavity size with a clear accumulation of cells (two out of two embryos), reminiscent of *ASPP2*<sup>ΔE4/ΔE4</sup> and *ASPP2*<sup>RAKA/RAKA</sup> mutant embryos' phenotype in a C57BL/6 background (Fig. 6e). Interestingly, the localisation pattern of

F-actin and Myosin at the apical surface of cells accumulating ectopically in *ASPP2*<sup>RAKA/RAKA</sup> mutant embryos was altered in a way similar to that observed at E7.5 in cells accumulating at the surface of the primitive streak. Together, this indicates that although *ASPP2*<sup>RAKA/RAKA</sup> mutants in a BALB/c background can bypass the proamniotic cavity phenotype, increasing mechanical stress is sufficient to make them again susceptible to it and suggests that ASPP2 may be required in response to increased mechanical stress to maintain epithelial tissue integrity.

**ASPP2 maintains the apical organisation of F-actin.** To understand how the absence of ASPP2 specifically regulates apical daughter cell reintegration into the epiblast, we analysed in detail its localisation pattern. ASPP2 was localised at the apical junctions in the VE (Fig. 7a and Supplementary Fig. 6a, b) and the epiblast (Fig. 7b and see Fig. 3a for antibody specificity). In the former, ASPP2 was uniformly distributed along the apical junctions forming a regular mesh at the surface of the embryo (Fig. 7a and Supplementary Fig. 6a, b). In the epiblast, however, ASPP2

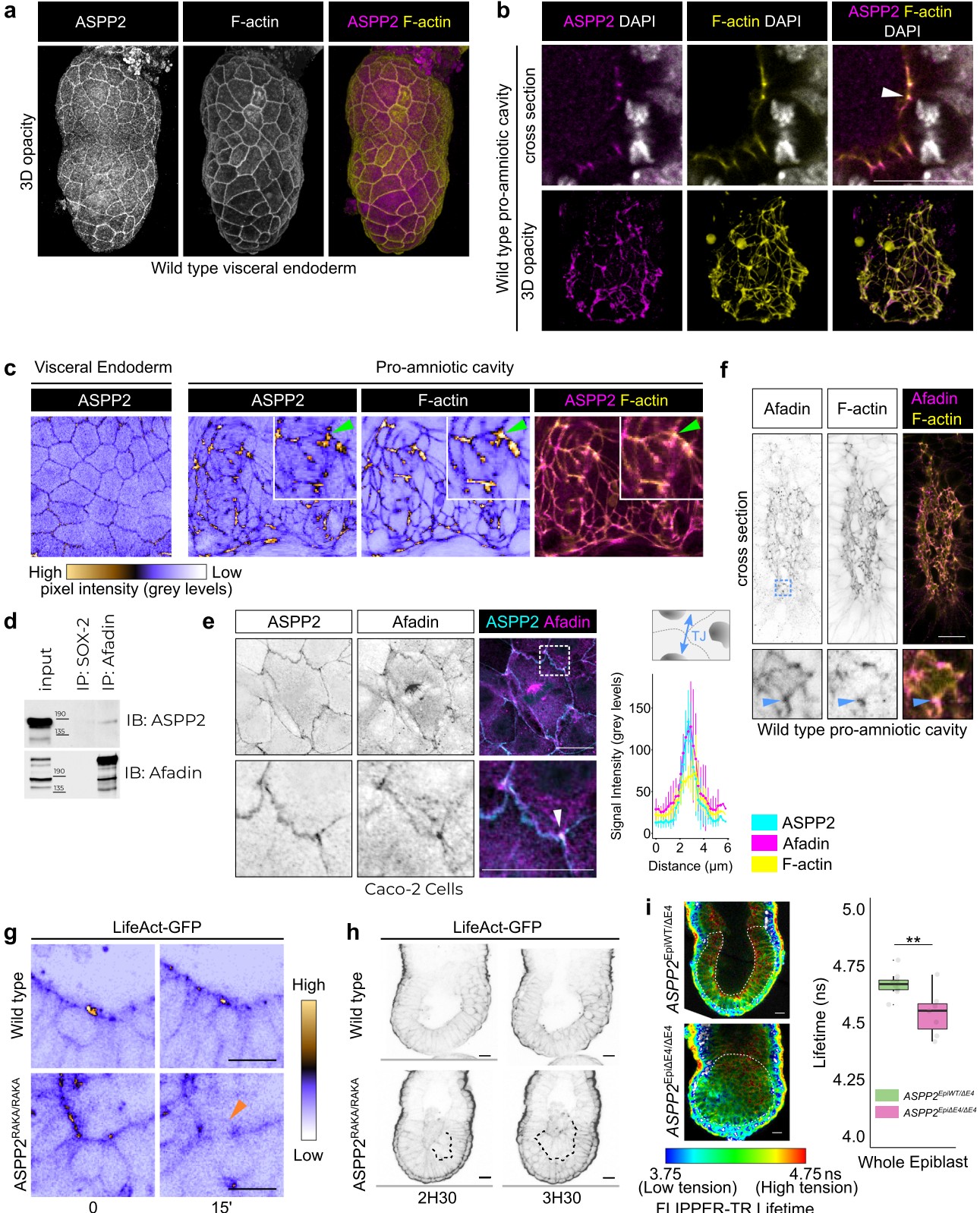

appeared enriched at specific locations along the apical junctions (Fig. 7b). The high curvature of the inner apical surface of the epiblast makes it difficult to examine from standard confocal volumes. We therefore computationally 'unwrapped'[47] the apical surfaces of the VE and epiblast so that we could more directly compare them (Fig. 7c). This revealed that, although ASPP2 was uniform in its distribution along all junctions in the VE, in the epiblast, it was enriched in specific locations, often coinciding with F-actin-rich tricellular junctions.

The enrichment of ASPP2 in regions of high F-actin and the disruption of F-actin localisation in mutants suggests that ASPP2 may somehow be linked to the F-actin cytoskeleton. Because ASPP2 does not possess any known F-actin binding domain, we looked if previously identified ASPP2 binding partners could

**Fig. 7 ASPP2 controls the localisation of apical F-actin and tensions in the epiblast. a** 3D opacity rendering showing the localisation of ASPP2 in E5.5 wild type embryos at the apical junctions of the visceral endoderm where it colocalises with F-actin. **b** Cross-section (top row) and 3D opacity rendering (bottom row) of the proamniotic cavity showing the localisation pattern of ASPP2 and F-actin at the apical junctions (white arrowhead). **c** The outer surface of the VE and apical surface of the epiblast were computationally 'unwrapped', revealing the enrichment of ASPP2 at specific locations along the apical junctions, often at F-actin-rich tricellular junctions (green arrowheads). **a–c** Representative images from six embryos. **d** The interaction between endogenous ASPP2 and the F-actin-binding protein Afadin was examined in Caco-2 cells by co-immunoprecipitation (representative images from three independent experiments). Molecular weights are indicated in kilodaltons. **e** The localisation pattern of endogenous ASPP2 and Afadin in Caco-2 cells was examined by immunofluorescence (representative images from five independent experiments). The bottom row represents the magnified region highlighted by a dotted box and shows the enrichment of ASPP2 and Afadin at tricellular junctions. ASPP2, Afadin and F-actin signal intensity was quantified across tricellular junctions (graph on the right). **f** The localisation pattern of Afadin in the proamniotic cavity was analysed by immunofluorescence in E6.5 wild type embryos. The blue arrowhead highlights the colocalisation of Afadin with F-actin at a tricellular junction. **g** The localisation pattern of F-actin was analysed by time-lapse microscopy in wild type (representative images from ten embryos) and $ASPP2^{RAKA/RAKA}$ (representative images from six embryos) LifeAct-GFP positive embryos. Note how apical F-actin is disrupted in $ASPP2^{RAKA/RAKA}$ LifeAct-GFP positive embryos (orange arrowhead). The colour scale represents pixel intensity (grey levels). **h** At later time points, the ectopic accumulation of cells in the epiblast of $ASPP2^{RAKA/RAKA}$ LifeAct-GFP positive embryos was evident (dotted line). **i** Representative FLIM images of $ASPP2^{EpiWT/\Delta E4}$ ($n = 9$ embryos) and $ASPP2^{Epi\Delta E4/\Delta E4}$ ($n = 7$ embryos) embryos (left) and comparison of mean lifetime values in the epiblast tissue, including delaminating cells (right). The dotted line highlights epiblast cells. For the box plots, the top and bottom lines of each box represent the 75th and 25th percentiles, respectively. The whiskers show the minima to the maxima values and the central line indicates the median. Outliers are represented with black dots. **p < 0.01 (unpaired two-sided Student's t-test, p = 4.84e-3). Nuclei and the F-actin cytoskeleton were visualised with DAPI and Phalloidin respectively. Scale bars: 20 µm. Source data are provided as a Source Data file.

provide this link. Interestingly, ASPP2 has been found to interact with Afadin in a number of proteomic studies[20,48]. Afadin is an F-actin-binding protein that has previously been shown to not only be enriched at tricellular junctions but also regulate their architecture[49]. Moreover, at E7.5, *Afdn*-null (*Afadin*-null) embryos display a phenotype reminiscent of the phenotype observed in E7.5 $ASPP2^{\Delta exon4}$ embryos (Supplementary Fig. 2b) with cells accumulating in the proamniotic cavity[50], suggesting that Afadin and ASPP2 have overlapping functions. To confirm that ASPP2 and Afadin can be found within the same protein complex, we immunoprecipitated endogenous Afadin in Caco-2 cells, a colorectal cancer cell line with strong epithelial characteristics that retains the ability to polarise. ASPP2 co-immunoprecipitated with Afadin, indicating that they are indeed found in the same protein complex (Fig. 7d). To further investigate where this complex might form, we analysed the localisation of endogenous ASPP2 and Afadin in Caco-2 and MDCK cells using super-resolution Airyscan microscopy. We found the proteins colocalised primarily at tricellular junctions, where F-actin was also enriched, including in dividing cells in metaphase (Fig. 7e and Supplementary Fig. 6c). Their expression pattern also partially overlapped at bicellular junctions (Supplementary Fig. 6c, d). Interestingly, Afadin was also found at the mitotic spindles (Fig. 7e) and cleavage furrow (Supplementary Fig. 6d), where ASPP2 was juxtaposed with Afadin. In E6.5 embryos, Afadin showed a similar localisation pattern to ASPP2, at the apical junction of cells in the epiblast and VE (Fig. 7f and Supplementary Fig. 6e).

Together, these results highlight the importance of the localisation pattern of ASPP2 in the epiblast, suggesting that it may be able to interact with F-actin at the apical junctions via its interaction with Afadin. To further investigate the role of ASPP2 in maintaining the organisation of the F-actin cytoskeleton in the epiblast, we generated $ASPP2^{RAKA/RAKA}$ embryos in a C57BL/6 background carrying a LifeAct-GFP transgene[51]. This allowed us to visualise F-actin in living embryos with time-lapse confocal microscopy (Fig. 7g, h and Supplementary Fig. 6f). In wild type embryos, we found that as epiblast cells divided, apical F-actin localisation was maintained (Fig. 7g). In contrast, in $ASPP2^{RAKA/RAKA}$ embryos, apical F-actin organisation was locally disrupted (Fig. 7g) and this was followed rapidly by the abnormal extrusion of cells into the proamniotic cavity (Fig. 7h). These results suggest that ASPP2 function is required to maintain the architecture of

apical F-actin in the epiblast and in its absence, actomyosin contractility at the apical junctions may be disrupted. If this is indeed the case, one might expect that the limiting membranes of cells accumulating in the proamniotic cavity of mutants would be under reduced tension. To test this possibility, we performed FLIPPER-TR lifetime measurements in E6.5 control and $ASPP2^{Epi\Delta E4/\Delta E4}$ embryos. In contrast to wild type embryos where regions of high tension delineated the proamniotic cavity, the epiblast of $ASPP2^{Epi\Delta E4/\Delta E4}$ embryos did not exhibit any organised pattern of tension (Fig. 7i and Supplementary Fig. 6g). Moreover, epiblast cells of $ASPP2^{Epi\Delta E4/\Delta E4}$ embryos as a whole exhibited significantly lower mean lifetimes (Mean = 4.54 ns, SD = 0.1 ns) in comparison to controls (Mean = 4.67 ns, SD = 0.053 ns), confirming that there is an overall reduction of membrane tension among epiblast cells in the absence of ASPP2 and organised apical F-actin. As with the abnormal apical daughter cell extrusion phenotype, this defect in membrane tension is specific to the epiblast, as the membrane tension at the apical surface of the exVE and emVE of control and $ASPP2^{Epi\Delta E4/\Delta E4}$ E6.5 embryos were comparable (Supplementary Fig. 6h, i). This suggests that in the pre-gastrulation embryo, ASPP2 is required to maintain the organisation of F-actin and tensions at the apical surface specifically of the epiblast.

**ASPP2 supports tissue integrity across a range of pseudostratified epithelia.** Next, we investigated whether ASPP2 was only required in the epiblast or whether, at later stages, it might also function in other tissues undergoing morphogenesis. At E7.5, during late primitive streak stages, mesoderm formation (marked by expression of T) and migration were broadly comparable between $ASPP2^{\Delta E4/\Delta E4}$ embryos and wild type littermates, despite the absence of a proamniotic cavity and the dramatic accumulation of cells now filling the entirety of the space inside the embryos (Fig. 8a and Supplementary Movie 3). Furthermore, there was no difference in the velocity, directionality and distance travelled by mesoderm cells migrating from wild type and $ASPP2^{\Delta E4/\Delta E4}$ mesoderm explants (Supplementary Fig. 7a and Supplementary Movie 4). This suggested that ASPP2 was not required for mesoderm specification or migration.

To test whether further patterning of the mesoderm occurred in the absence of ASPP2, we examined $ASPP2^{\Delta E4/\Delta E4}$ embryos at E8.5. Morphologically, these embryos were severely disrupted, shorter

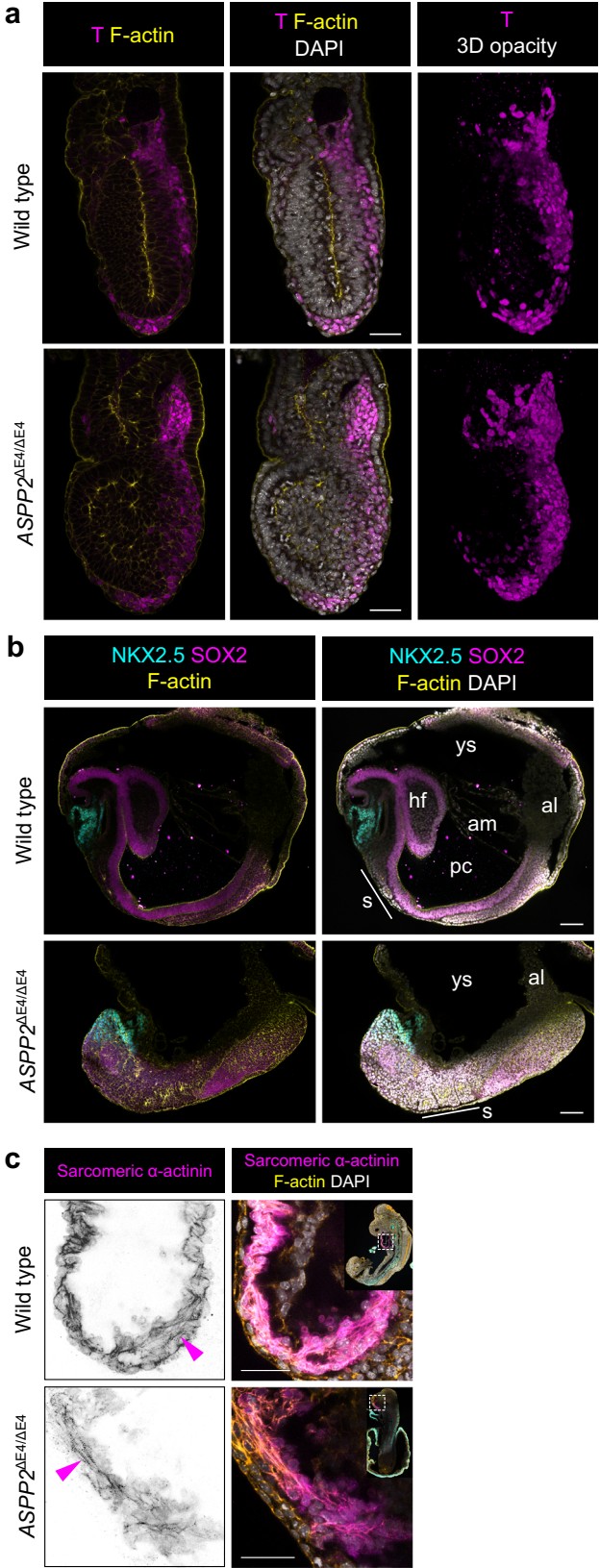

**Fig. 8 ASPP2 is not required for cell fate specification during gastrulation. a** The primitive streak expands comparatively in E7.5 wild type and ASPP2$^{\Delta E4/\Delta E4}$ embryos. Mesoderm cells were labelled by immunofluorescence using an antibody against Brachyury (T). **b** Patterning proceeds normally in the absence of ASPP2. The ectoderm and cardiac progenitors were labelled in E8.5 wild type and ASPP2$^{\Delta E4/\Delta E4}$ embryos with antibodies against SOX2 and NKX2.5, respectively. ys yolk sack, al allantois, s somites, hf head fold, am amnion, pc proamniotic cavity. **c** Cardiac progenitors can differentiate into cardiomyocytes in E9.5 ASPP2$^{\Delta E4/\Delta E4}$ embryos. The presence of the contractile machinery (magenta arrowheads) was assessed in wild type and ASPP2$^{\Delta E4/\Delta E4}$ embryos using an antibody against sarcomeric α-actinin. Nuclei and the F-actin cytoskeleton were visualised with DAPI and Phalloidin, respectively. Scale bars: 50 μm (**a**, **c**), 100 μm (**b**).

also positive for sarcomeric α-actinin, suggesting that the NKX2-5 positive cells could differentiate into cardiomyocytes (Fig. 8c and Supplementary Fig. 7b). Next, we analysed whether the mesoderm could go on to form structurally normal somites. Using FOXC2 as a marker of somitic mesoderm, we were able to identify distinct somite-like structures in ASPP2$^{\Delta E4/\Delta E4}$ embryos (Fig. 9a and Supplementary Fig. 7b). However, we found that these somites were smaller than normal (Fig. 9b). Compared to wild type control somites, a large proportion of mutant somites exhibited disrupted epithelial organisation (Fig. 9a, c) and a reduced proportion of formed cavities (Fig. 9a, d). The relative size of the somitocoel in ASPP2$^{\Delta E4/\Delta E4}$ embryos was also significantly reduced in comparison to wild type controls (Fig. 9e). Importantly, somites with disrupted epithelial organisation and lacking cavities had features reminiscent of ASPP2$^{\Delta E4/\Delta E4}$ epiblasts: cells could be seen accumulating in the centre resulting in the obliteration of the somitocoel (Fig. 9a). These cells also displayed a lack of apical Par6, suggesting that as in the epiblast, in the forming somites, apical-basal polarity was disrupted (Fig. 9f).

These results suggest that ASPP2 is required not only in the epiblast, but more generally in pseudostratified epithelia[15]. To investigate this further, we tested the requirement for ASPP2 in lumen formation during cystogenesis. We derived embryonic stem cells (ESC) from embryos with exon 4 of ASPP2 flanked by two LoxP sites. To generate ASPP2$^{\Delta E4/\Delta E4}$ ESC, they were infected with a CRE-recombinase-expressing adenovirus. When grown in Matrigel, we found that the majority of ASPP2$^{\Delta E4/\Delta E4}$ ESC-derived cysts failed to form lumens in comparison to control cysts (Supplementary Fig. 7c). Similarly, ESC derived from ASPP2$^{RAKA/RAKA}$ embryos failed to form lumens in comparison to wild type ESC, suggesting that the formation of lumens during cystogenesis requires ASPP2/PP1 interaction (Supplementary Fig. 7d).

Since our data suggest that ASPP2 is required in regions undergoing increased mechanical stress (Fig. 6), we wanted to examine further the potential importance of ASPP2 during head fold formation. We took advantage of the ASPP2$^{RAKA/RAKA}$ embryos in a BALB/c background as their phenotype is milder and they develop a proamniotic cavity (Fig. 5a), reducing the likelihood that phenotypes observed in the head fold region are secondary defects due to overall tissue disorganisation. At E8.5, wild type embryos exhibited fully formed head folds (Fig. 9g). In contrast, ASPP2$^{RAKA/RAKA}$ embryos failed to form head folds in the rostral region of the ectoderm. In this region, the epithelium buckled locally, without being able to fully complete head fold morphogenesis. Interestingly, this was accompanied by a loss of organisation of F-actin at the apical junction, similarly to what was observed in the proamniotic cavity of ASPP2$^{\Delta E4/\Delta E4}$ embryos and in the primitive streak of ASPP2$^{RAKA/RAKA}$ embryos in a

along the anterior-posterior axis and without head folds (Fig. 8b). However, using the cardiac progenitor marker NKX2.5, we found that this population of cells was able to migrate rostrally despite the dramatic morphological defects present in ASPP2$^{\Delta E4/\Delta E4}$ embryos (Fig. 8b). At E9.5, some cells in the anterior of these embryos were

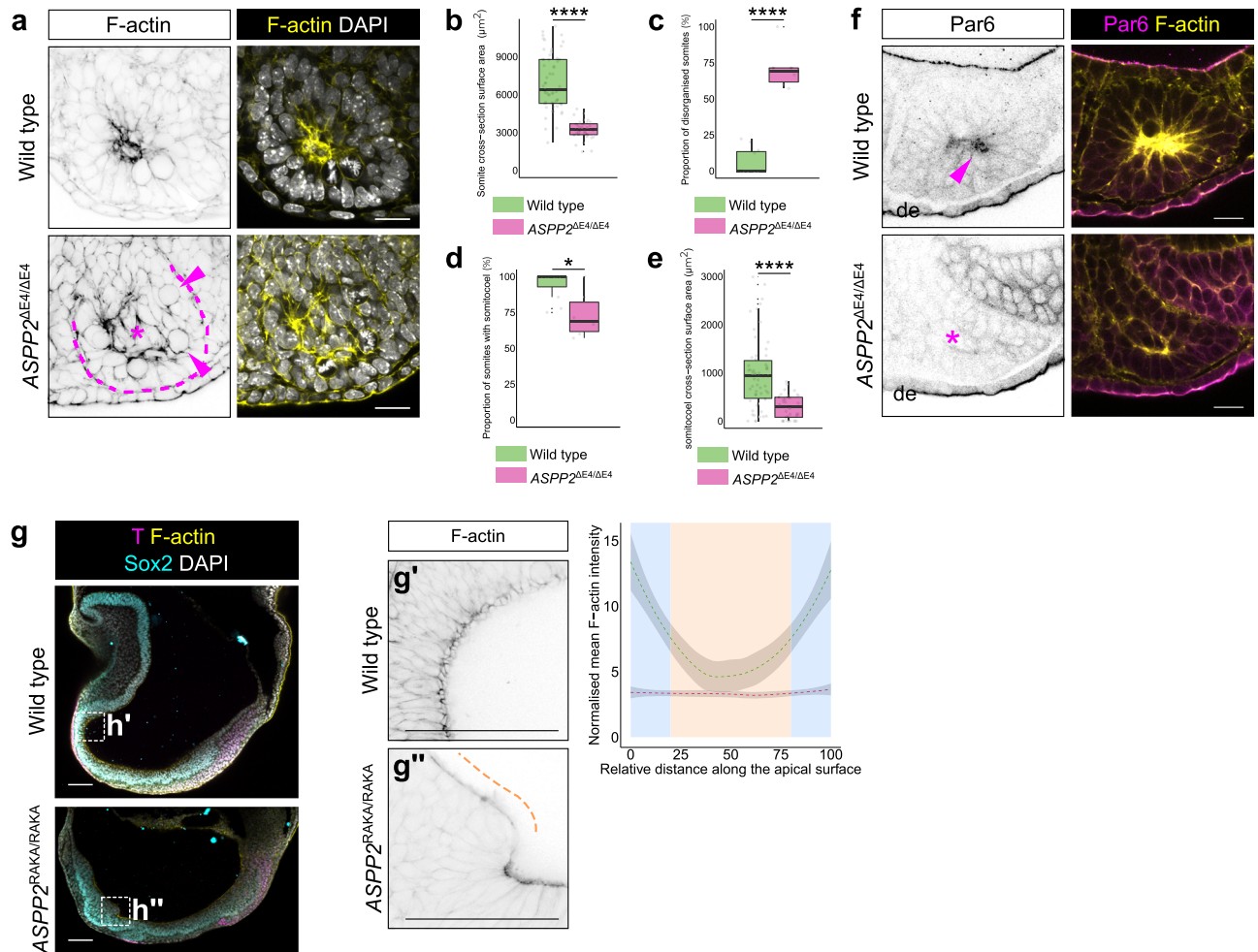

**Fig. 9 ASPP2 is required for tissue integrity across a variety of pseudostratified epithelia. a** Somite architecture is disrupted in $ASPP2^{\Delta E4/\Delta E4}$ embryos. The dotted line highlights the contour of a somite in an $ASPP2^{\Delta E4/\Delta E4}$ embryo. The star indicates the ectopic accumulation of cells in the centre of this somite. Arrowheads point to mitotic figures. **b–e** Quantification of somite characteristics in wild type ($n = 10$ embryos, 58 somites) and $ASPP2^{\Delta E4/\Delta E4}$ ($n = 6$ embryos, 35 somites) embryos at E8.5. For the box plots, the top and bottom lines of each box represent the 75th and 25th percentiles, respectively. The whiskers show the minima to the maxima values and the central line indicates the median. Outliers are represented with black dots. *$p < 0.05$, ****$p < 0.0001$ (unpaired two-sided Student's $t$-test; $p = 0$ in **b**, $p = 3.093e-05$ in **c**, $p = 0.025$ in **d**, $p = 1.116e-08$ in **e**). **f** Apical-basal polarity is defective in the somites of $ASPP2^{\Delta E4/\Delta E4}$ embryos. Par6 localised apically in wild type somites (arrowhead) whereas it was absent in $ASPP2^{\Delta E4/\Delta E4}$ embryos (star). de definitive endoderm. **g** Head fold formation is defective in $ASPP2^{RAKA/RAKA}$ embryos. The organisation of apical F-actin was disorganised locally in the anterior ectoderm of $ASPP2^{RAKA/RAKA}$ embryos (orange dotted line). F-actin signal intensity along the apical surface of ectoderm cells in disrupted areas in $ASPP2^{RAKA/RAKA}$ embryos ($n = 3$ embryos, five cells per embryo) was compared to wild type cells ($n = 3$ embryos, five cells per embryo). Measurements were made on cross-sections along the apical domain of individual ectoderm cells from apical junction to the apical junction (represented with a blue background in the graph). The 95% confidence interval is represented by the grey area. Nuclei and the F-actin cytoskeleton were visualised with DAPI and Phalloidin, respectively. Scale bars: 20 μm (**a**, **f**), 100 μm (**g**). Source data are provided as a Source Data file.

BALB/C background (Fig. 9g). These results strongly suggest that ASPP2 is required to maintain tissue integrity by regulating F-actin organisation at the apical junctions as tensions increase in the rostral region of the ectoderm during head fold formation. Together, these results also reinforce the idea that ASPP2, and its interaction with PP1, are required during morphogenetic events that result in increased tensions at the level of apical junctions in epithelial tissues.

## Discussion

Our study unveils a central role for ASPP2 in maintaining the integrity of pseudostratified epithelia under increased mechanical stress during major morphogenetic events: in the forming proamniotic cavity, in the primitive streak during gastrulation, in somites and in the head fold region. Specifically, we show that ASPP2 is required in the epiblast at E6.5 where apical tensions are

higher. We also show that, in the absence of functional ASPP2 proteins, increased mechanical stress can disturb epithelial structural integrity in tissues that are normally aphenotypic.

Though it is required across a diverse range of epithelia, ASPP2 is not required in all epithelia it is expressed during early mouse embryonic development. Instead, it is specifically required in pseudostratified epithelia and in particular, at the egg cylinder stage, in the epiblast. Our results highlight crucial differences between the epiblast, where ASPP2 is required and another epithelial tissue at this stage, the VE, where it is not required: aside from the former being a pseudostratified and the latter a simple epithelium, apical tensions are higher in the epiblast, and the apical junctions of its cells differ in comparison to those of the VE. For example, E-cadherin and SCRIB, proteins that are normally localised predominantly basolaterally in other tissues, are enriched closer to the apical junctions in the epiblast, potentially as additional support to

withstand higher tensions and maintain tissue cohesion. The enrichment of ASPP2 at tri- or multi-cellular junctions in the epiblast is consistent with this tissue experiencing increased apical tensions. Moreover, the overall configuration of pseudostratified epithelia—high cellular density and cells with narrow apical domains—is likely to contribute to the impact of cell rounding during mitosis on tensions exerted at the level of apical junctions[2]. Finally, SHROOM2 is also enriched at the apical junctions specifically in the epiblast, suggesting that apical constriction is higher in this tissue. This agrees with apical tension being higher in cells of the epiblast and is likely required to maintain their narrow apical domain and the curvature of the proamniotic cavity. These observations, together with the requirement for ASPP2 in the primitive streak of ASPP2[RAKA/RAKA] embryos, where apical constriction also plays an important role, suggest that ASPP2 might be required specifically when cells actively constrict apically and are subject to increased mechanical stress.

StriPerhaps, ASPP2 is required for the maintenance of proamniotic cavity architecture via a mechanism that, following cell divisions, specifically prevents the most apical daughter cells from delaminating apically. ASPP2 achieves this by maintaining the integrity and organisation of the F-actin cytoskeleton at the apical surface of dividing cells. This mechanism is consistent with ASPP2 playing a role in maintaining epithelial integrity under increased mechanical stress, given that mitotic rounding results in forces sufficient to contract the epithelium in the apical-basal axis and mechanically contribute to the expansion of the lumen[2]. However, it remains unclear why it is only the more apically localised daughter cells that are affected in ASPP2 mutant embryos. One possibility is that these daughters do not inherit the basal process that tethers cells to the basement membrane in pseudostratified epithelia and rely on intact apical F-actin organisation and tensions to reintegrate basally. However, the idea that the basal process is inherited asymmetrically is controversial[52]. Interestingly, a study in the epidermis suggests that some apical daughters retain a basal endfoot that enables the reorientation of cell divisions within the plane of the epithelium. Moreover, this study shows that Afadin functions in endfoot retention[53]. We could not clearly establish whether daughter cells were making contact via a basal process in our live imaging experiments mainly because of the high cellular density of the epiblast. However, considering the link between ASPP2 and Afadin, it will be interesting to test this hypothesis further using, for example, mosaically-labelled epiblast cells. Studies in Drosophila have suggested there exist alternative mechanisms involving lateral adhesion mediated by immunoglobulin superfamily cell adhesion molecules that support daughter cell reintegration[54,55]. Our data suggest that the apical junctions may play an equivalent role in the epiblast and more generally in pseudostratified epithelia.

It has been suggested that the purpose of interkinetic nuclear migration is to ensure that cells divide apically to safeguard the integrity of pseudostratified epithelia[56]. Here we show that this is not sufficient to maintain tissue integrity as, in the absence of ASPP2, the apical-basal movement of nuclei and the position of cell divisions during IKNM proceeds unhindered. Our observations also suggest that the organisation of the F-actin cytoskeleton at the apical junctions is not required for nuclear movement during INKM and that intact basolateral domain and attachment to the basement membrane are sufficient instead.

Given that the ASPP2 interactors Par3[57,58] and Afadin[59–61], tricellular junctions[62], as well as tissue tension[63–66] have all been shown to determine cell division orientation to some extent, it was important to explore whether ASPP2 could play a role in this process. Our results indicate that ASPP2 does not control the orientation of cell divisions in the epiblast. Similar to previous

work[42], we also find no bias towards cell divisions orientated in the plane of the epithelium, suggesting that, at these early stages of development, planar cell polarity may not play a role in directing cell division orientation. We however cannot rule out that, in a different context, ASPP2 might control cell division orientation in conjunction with Afadin. In fact, later in development in E8.5 ASPP2[RAKA/RAKA] embryos, cells sometimes delaminated basally in the anterior regions of the ectoderm, reminiscent of the phenotype observed in SCRIB- and DLG-depleted Drosophila wing discs, where cell divisions are normally orientated in the plane of the epithelium by cell-cell junctions to maintain epithelial integrity[67].

Our results highlight the previously underappreciated discrete localisation pattern of ASPP2 along the apical junctions in epithelial cells, in particular of the epiblast, where it resembles that of Afadin. Our results also reveal the function of the PP1-binding site of ASPP2 in the regulation of F-actin organisation at the apical junction. There are many interesting overlaps between ASPP- and Afadin-related phenotypes, particularly from work in Drosophila, supporting the idea that they work in a common pathway involved in the regulation of F-actin, contractility, cell shape and epithelial tissue organisation. For example, experimentally perturbing the activity of Cno (Drosophila Afadin)[68] and dASPP[29] commonly results in the alteration of pigment cell shape and organisation in the developing pupal retina. Importantly, in the retina, dASPP also promotes the junctional localisation of specific PP1 isoforms, suggesting that the recruitment of PP1 by dASPP is essential to its function[32]. In the mouse, Afadin, is required not only for lumen morphogenesis in the epiblast[50] similar to ASPP2, but also in developing renal tubes[60,69] and pancreas[70]. Moreover, ASPP2 contributes to mesenchymal to epithelial transition in mouse kidney in vivo[71]. It will therefore be interesting to investigate further whether ASPP2 is also required in the developing kidney or pancreas using tissue-specific ablation of its function.

Considering that Afadin regulates the architecture of tricellular junctions in response to tensions[49,72], the interaction between Afadin and ASPP2 strongly suggests that ASPP2 may exert its F-actin regulatory function at tricellular junctions via Afadin. The role of Afadin in regulating the linkage between F-actin and junctions during apical constrictions[73] suggests that ASPP2 may also be important in this process, which may be particularly relevant in the primitive streak. Tricellular junction are emerging as a particularly important aspect of tissue homoeostasis, at the intersection between actomyosin contractility and apical-basal organisation in the context of tissue tensions[74]. It will therefore be important to test whether ASPP2 is directly involved in the response to tissue tension by interacting with Afadin at the level of tricellular junctions to maintain F-actin organisation.

Our study also suggests that the interaction between ASPP2 and PP1 might be essential to the well-documented tumour suppressor function of ASPP2[23,71,75]. Simply abrogating the ability of ASPP2 to recruit PP1 is enough to induce the formation of abnormal discrete clusters of cells in the epiblast reminiscent of tumours. This suggests that mutations in ASPP2 that interfere with its interaction with PP1 might, in conjunction with mechanical stress, lead to tumour development. These mutations could be in the canonical PP1-binding domain of ASPP2, but also in other key domains which have been shown to contribute to the interaction[32]. Recent findings support the idea that ASPP2 mutations could lead to tumorigenesis in the presence of mechanical stress. Using insertional mutagenesis in mice with mammary-specific inactivation of Cdh1, ASPP2 was identified as part of a mutually exclusive group containing three other potential tumour suppressor genes (Myh9, Ppp1r12a and Ppp1r12b), suggesting that these genes target the same process[76].

With our finding that ASPP2 controls the organisation of the F-actin cytoskeleton, it now becomes apparent that, in addition to three of these genes being PP1-regulatory subunits, all four are in fact F-actin regulators. Biological studies to test specific mutations found in ASPP2 in cancer and elucidating the substrates and specific phospho-residues targeted by the ASPP2/PP1 complex will therefore provide new insights into the tumour suppressor role of ASPP2 and might help develop new approaches to cancer treatment.

## Methods

**Mouse strains and embryo generation.** All animal experiments complied with the UK Animals (Scientific Procedures) Act 1986, were approved by the local Biological Services Ethical Review Process and were performed under UK Home Office project licences PPL 30/3420 and PCB8EF1B4. The LERP (local ethical review panel) at the Department of Physiology, Anatomy and Genetics approved the study.

All mice were maintained on a 12-h light, 12-h dark cycle, with a room temperature of 19–23 °C and 45–65% humidity. Noon on the day of finding a vaginal plug was designated 0.5 dpc. For preimplantation stages, embryos were flushed using an M2 medium (Sigma M7167) at the indicated stages. For post-implantation stages, embryos of the appropriate stage were dissected in an M2 medium with fine forceps and tungsten needles.

We originally obtained *ASPP2* mutant mice in which exons 10–17 were replaced with a neo-r gene[40] from Jackson Laboratory. After careful characterisation of this mouse line, we found that the Neo cassette was not inserted in the *ASPP2* locus. As a consequence, we used a different strategy to generate *ASPP2* mutant mice. C57BL/6N-Trp53bp2<tmIa>(EUCOMM) heterozygous sperm (obtained from the Mary Lyon Centre) was initially used to fertilise ACTB:FLPe B6J homozygous oocytes (Jackson Laboratory). This resulted in the removal by the flippase of the LacZ and neo-r region flanked by FRT sites and the generation of heterozygous mice with one allele of *ASPP2* in which exon 4 was flanked by LoxP sites. Those mice were bred in a C57BL/6J background for over four generations to breed out the rd8 mutation in the *CRB1* gene found in the C57BL/6 N background and eliminate the remaining FRT site left behind. They were then crossed to generate mice homozygous for the *ASPP2* conditional allele in a C57BL/6 J background (*ASPP2*flE4/flE4 mice). These mice were also crossed with *Sox2Cre* mice[77] to generate mice with Exon 4 excised in one allele of *ASPP2* (*ASPP2*WT/ΔE4 mice). *ASPP2*WT/ΔE4 mice were subsequently backcrossed into wild type C57BL/6 J mice to segregate out the *Sox2Cre* transgene.

*ASPP2*WT/ΔE4 mice were used to generate *ASPP2*ΔE4/ΔE4 embryos. To produce epiblast-specific *ASPP2*-null embryos (*ASPP2*EpiΔE4/ΔE4 embryos), *ASPP2*WT/ΔE4 mice homozygous for the *Sox2Cre* transgene were crossed with *ASPP2*flE4/flE4 mice. To generate *ASPP2*ΔE4/ΔE4 embryos with fluorescently labelled membranes, we established *ASPP2*WT/ΔE4 mice homozygous for the mT/mG transgene[78] and crossed them with *ASPP2*WT/ΔE4 mice.

The *ASPP2*WT/RAKA mice were made by inGenious Targeting Labs (Ronkonkoma, NY). A BAC clone containing exon 14 of the *trp53bp2* gene was subcloned into a ~2.4 kb backbone vector (pSP72, Promega) containing an ampicillin selection cassette of the construct prior to retransformation prior to electroporation. A pGK-gb2 FRT Neo cassette was inserted into the gene. In the targeting vector, the wild type GTG AAA TTC was mutated to GCG AAA GCC by overlap extension PCR and introduced into C57BL/6 × 129/SvEv ES cells by electroporation. Inclusion of the mutations in positive ES cell clones was confirmed by PCR, sequencing and Southern blotting. ES cells were microinjected into C57BL/6 blastocysts and resulting chimeras mated with C57BL/6 FLP mice to remove the Neo cassette. The presence of the mutation was confirmed by sequencing. Mice were then backcrossed with BALB/cOlaHsd or C57BL/6 J mice for at least eight generations to obtain the RAKA mutation in the respective pure background. *ASPP2*RAKA/RAKA embryos were generated from heterozygous crosses. To generate LifeAct-GFP-positive *ASPP2*RAKA/RAKA embryos, we generated *ASPP2*WT/RAKA mice heterozygous for the LifeAct-GFP transgene[51].

**siRNA microinjections.** siGENOME RISC-Free Control siRNA (Dharmacon) and Silencer Select Pre-designed siRNAs against mouse ASPP2 (#4390771, siRNA s102092, Ambion) were resuspended in nuclease-free sterile water and used at 20 μM. For zygotes, 3 to 4-week-old CD-1 females (Charles River UK) were injected intraperitoneally with 5 IU of PMSG (Intervet) and 48 h later with 5 IU of hCG (Intervet), and were paired with C57Bl/6 J male mice (in house). Zygotes were retrieved from oviductal ampullae at 20 h post-hCG. Cumulus-enclosed zygotes were denuded by exposure to 1 mg/mL hyaluronidase (Sigma) in modified mHTF (Life Global) containing 3 mg/ml BSA for 3–6 min and cultured in LGGG-020 (life Global) containing 3 mg/ml BSA in the presence of 5% CO₂ at 37 °C. Micro-injection of zygotes commenced 2 h after release from cumulus mass. Zygotes with normal morphology were microinjected into the cytoplasm in 30 μl drops of modified HTF media containing 4 mg/ml BSA using a PMM-150FU Piezo impact drive (Primetech) using homemade glass capillaries with ~5–10 pl of siRNA.

Zygotes were returned to LGGG-020 containing 3 mg/ml BSA in the presence of 5% CO₂ at 37 °C until analysis.

**Human embryo collection.** Human embryos were donated from patients attending the Oxford Fertility with approval from the Human Fertilization and Embryology Authority (centre 0035, project RO198) and the Oxfordshire Research Ethics Committee (NRES Committee South Central—Berkshire B; Reference number 14/SC/0011). Informed consent was attained from all patients. The study design and conduct complied with all relevant regulations regarding the use of human study participants and was conducted in accordance with the criteria set by the Declaration of Helsinki. All new patients intending to come to the unit for fertility treatment were given an information pack when they attended the evening meeting before starting treatment. An Information sheet about research projects using surplus eggs and embryos was included in the pack. Patients would not typically visit the clinic until several weeks after receiving this, giving time for them to consider whether or not they want to participate. All patients commencing their fertility treatment then arranged a routine new patient consultation appointment. At this visit doctors/nurses would check that the patient meets the inclusion criteria to participate in the study. This includes checking that the patient has, in a questionnaire supplied to ALL patients by the HFEA (Form WT), agreed in principle to being approached about research projects involving their gametes (eggs). If so, they would ask the patient if they wanted to participate in the study. A research nurse would always be available for further discussion of the projects if necessary. There was no patient compensation. Embryos were fixed in 4% paraformaldehyde, washed twice and kept in PBS containing 2% bovine serum albumin (PBS-BSA) at 4 °C until they were used for immunohistochemistry.

**Wholemount immunohistochemistry.** Post-implantation embryos were fixed in 4% paraformaldehyde in phosphate-buffered saline (PBS) at room temperature for 20 to 45 min depending on embryo stages. Embryos were washed twice for 10 min in 0.1% PBS-Tween (PBS containing 0.1% Tween 20). Embryos were then permeabilized with 0.25% PBS-Triton (PBS containing 0.25 Triton X-100) for 25 min to 1 h depending on embryo stages and then washed twice for 10 min in 0.1% PBS-Tween. Embryos were incubated overnight in a blocking solution (3% bovine serum albumin, 2.5% donkey serum in 0.1% PBS-Tween). The next day, primary antibodies were diluted in blocking solution and added to the embryos overnight. The following day, embryos were washed three times for 15 min in 0.1% PBS-Tween and then incubated with secondary antibodies and Phalloidin diluted in blocking solution overnight. Finally, embryos were washed four times in 0.1% PBS-Tween and kept in DAPI-containing VECTASHIELD Antifade Mounting Medium (Vector Laboratories) at 4 °C until used for imaging. Short incubation steps were carried out in wells of a 12-well plate on a rocker at room temperature and overnight steps were carried out in 1.5 ml Eppendorf tubes at 4 °C.

For preimplantation embryos, fixation and permeabilization times were reduced to 15 min and 2% PBS-BSA (PBS containing 2% bovine serum albumin) was used for washing steps. Blocking and secondary antibody incubation steps were reduced to one hour. Embryos were transferred between solutions by mouth-pipetting. The embryos were mounted in eight-well chambers in droplets consisting of 0.5 μl DAPI-containing VECTASHIELD and 0.5 μl 2% PBS-BSA. After mounting the embryos were kept in the dark at 4 °C until they were imaged.

**Immunocytochemistry.** Caco-2 and MDCK cells were maintained in Dulbecco's modified Eagle's medium containing 10% foetal bovine serum, penicillin, and streptomycin at 37 °C in a 5% CO₂ atmosphere incubator. In preparation for immunocytochemistry, Caco-2 cells were seeded onto coverslips in 24-well plates with fresh medium. Forty-eight hours later, cells were fixed with 4% paraformaldehyde (in PBS) for 10 min, washed twice in PBS and then permeabilized with 0.1% Triton X-100 in PBS for 4 min. Cells were washed twice in PBS and 2% PBS-BSA was then used as a blocking solution for 30 min prior to incubation with primary antibodies. Primary antibodies were diluted in 2% PBS-BSA and applied to cells for 40 min. Cells were then washed three times with PBS. Secondary antibodies (1:400), DAPI (1:2000, Invitrogen) and Phalloidin (1:400) were diluted in 2% PBS-BSA and applied to cells for 20 min. Coverslips were then washed three times with PBS and mounted onto glass slides with a small drop of Fluoromout-G (SouthernBiotech). They were air-dried before being sealed with nail varnish. All incubation steps were carried out at room temperature on a rocker. Samples were kept in the dark at 4 °C until they were imaged.

**Antibodies and phalloidin conjugates.** The following antibodies were used at the stated dilutions: rabbit anti-ASPP2 (Sigma, HPA021603), 1:100-1:200 (IHC); mouse anti-ASPP2 (Santa Cruz Biotechnologies, sc135818), 1:100 (ICC), 1:1000 (IB); mouse anti-YAP (Santa Cruz Biotechnology, sc-101199), 1:100 (IHC); rabbit anti-pYAP S127(Cell Signaling, 4911), 1:100 (IHC); rabbit anti-Par3 (Millipore, 07-330), 1:100 (IHC); rabbit anti-Pard6b (Santa Cruz Biotechnology, sc-67393), 1:100 (IHC); rabbit anti-SCRIB (Santa Cruz Biotechnology, sc28737), 1:100 (IHC); rat anti-E-cadherin (Sigma, U3254), 1:100 (IHC); goat anti-SOX17 (R&D Systems, AF1924), 1:100 (IHC); rabbit anti-Phospho-Histone H3 (Cell Signaling, 9713), 1:200 (IHC); rabbit anti-Cleaved Caspase-3 (Cell Signaling, 9661), 1:100 (IHC); goat anti-Brachyury (Santa Cruz Biotechnology, sc17745), 1:100 (IHC); rabbit anti-

Sarcomeric α-actinin (Abcam, ab68167), 1:100 (IHC); mouse anti-FOXC2 (Santa Cruz Biotechnology, sc515234), 1:100 (IHC); rabbit anti-SOX-2 (Millipore, AB5603), 2 μl per mg of cell lysate (co-IP), 1:100 (IHC); goat anti-NKX2.5 (Santa Cruz Biotechnology, sc8697), 1:100 (IHC); rabbit anti-Afadin (Sigma, A0224), 2 μl per mg of cell lysate (co-IP), 1:100 (IHC, ICC), 1:1000 (IB); rabbit anti-Laminin (Sigma, L9393), 1:200 (IHC); goat anti-AMOT (Santa Cruz Biotechnologies, sc82491), 1:200 (IHC); goat anti-GATA-6 (R&D Systems, AF1700), 1:100 (IHC); rabbit anti-Myosin IIa (Cell Signaling, #3403), 1:100; rabbit anti-phospho-Myosin light chain 2 (Cell Signaling, #3674), 1:100. The following were used at 1:100 for IHC and 1:400 for ICC: Alexa fluor 555 donkey-anti-mouse (Invitrogen, A-31570), Alexa fluor 647 goat anti-rat (Invitrogen, A-21247), Alexa fluor 488 donkey-anti-rabbit (Invitrogen, A21206), Phalloidin-Atto 488 (Sigma, 49409), Phalloidin–Atto 647 N (Sigma, 65906).

**Confocal microscopy, image analysis and quantification**. Samples were imaged on a Zeiss Airyscan LSM 880 confocal microscope with a C-Apochromat 40x/1.2 W Korr M27 water immersion objective or a Plan-Apochromat 63x/1.4 OIL DIC M27 objective. For super-resolution imaging, an Airyscan detector was used[79]. Volocity (version 6.3.1, PerkinElmer) and Zen (Zeiss) software were used to produce maximum intensity projections and 3D opacity renderings. Image analysis was performed on optical sections. For signal intensity profiles along the apical-basal axis and across tricellular junctions, the arrow tool in the Zen software was used. Anterior and posterior embryo widths measurements were made using the line tool in Volocity.

For F-actin signal intensity profiles across the apical surface of epiblast or ectoderm cells, Fiji's freehand line tool with a width of '3' was used[80]. Because the size of the apical domain was different for each cell measured, distances were expressed as percentages, with 100% representing the total distance across the apical domain. To account for depth-dependent signal attenuation, F-actin signal intensity at the apical domain was normalised by mean F-actin intensity in the nucleus of the cell measured. In each experiment, for each genotype, three embryos were used for measurements and five cells were analysed per embryo. The LOWES method was used to fit a line to the data.

**Mouse embryo culture for live imaging and image analysis**. To restrain embryo movement during imaging, lanes were constructed inside the eight-well Lab-Tek II chamber slide (Nunc), using glass rods made from hand-drawn glass capillaries. Shorter pieces were used as spaces between two rods to create a space slightly wider than an embryo. Silicone grease was used to maintain the rods together. Each well was filled with medium containing 50% phenol red-free CMRL (PAN-Biotech, Germany) supplemented with 10 mM L/glutamine (Sigma-Aldrich) and 50% Knockout Serum Replacement (Life Technologies, England). The chamber was equilibrated at 37 °C and an atmosphere of 5% $CO_2$ for at least 2 h prior to use. Freshly dissected embryos were placed in the lanes between two rods and allowed to settle prior to imaging on a Zeiss LSM 880 confocal microscope equipped with an environmental chamber to maintain conditions of 37 °C and 5% $CO_2$. Embryos were imaged with a C-Apochromat 40x/1.2 W Korr M27 water immersion objective. Using a laser excitation wavelength of 561 nm, embryos labelled with mT/mG were imaged every 7.5 min and for each time point, nine z-sections were acquired every 3 μm around the midsagittal plane for up to 10 h. For LifeAct-positive embryos, a laser excitation wavelength of 488 nm was used, and embryos were imaged every 15 min for 6 h. For each time point, 12 z-sections every 1.5 μm were collected around the midsagittal plane.

Daughter cell movement was quantified using the Fiji plugin TrackMate (v5.2.0)[81]. Timepoints were registered using Fiji. The jittering was accounted for by correcting cell coordinates relatively to the centre of the embryonic region. The distance travelled by daughter cells (d) was analysed by calculating the distance between the coordinates of their final position and the coordinates of their respective mother cell immediately prior to cell division. The direction of daughter cell movement (θ) was analysed by calculating the angle between the vector describing cell movement (that is the vector originating from the coordinates of the mother cell immediately prior to cell division to the coordinates of the daughter cell at its final position) and the vector from the coordinates of the mother cell prior cell division to the coordinates of the embryonic region's centre. To establish the angle of cell division, we first defined a vector starting at the coordinates of one daughter and ending at the coordinates of the other immediately after cell division. We then defined a second vector originating halfway between the two daughters and terminating at the centre of the embryonic region. The angle of cell division was defined as the angle between those two vectors. The relative position of cell divisions was defined as the distance between the position of the mother cell immediately prior to cell division and the base of the epiblast.

**FLIM measurements and analysis**. For FLIM measurements, embryos were cultured in phenol red-free M2 and incubated for 2 h with FLIPPER-TR probe (Spirochrome) at 1 μM prior to imaging. The labelled embryos were imaged in eight-well glass-bottom chambers slides (#1.5 glass, ThermoFisher Scientific) and all images were acquired in the midsagittal plane. The FLIM measurements were performed on a Leica SP8 equipped with a Fast Lifetime Contrast (FALCON) module allowing for quick acquisitions at high photon counts[82]. The embryos were

imaged at 37 °C using a 20x multi-immersion objective (Leica C PL APO CS2 20x/0.75 IMM) using water as an immersion medium. The FLIPPER-TR fluorescence was excited at 488 nm with a tuneable white light laser (WLL; NKT Photonics) pulsing at a 20 MHz repetition rate accommodating for the relatively long lifetime decay of the probe. The master power of the WLL was set to 70% and we used 5–10% of that for excitation (corresponding to around 50 μW). The zoom was set to 1.5 yielding a 387.63 μm$^2$ × 387.63 μm$^2$ field of view covered at Nyquist by 2936 × 2936 pixels$^2$ (pixel size 135 nm/px). The pinhole was set to 1.2 AU, scan speed was 200 Hz and 25 images were accumulated. Fluorescence was collected in a window from 499 to 701 nm on an internal HyD-SMD detector (Leica Microsystems). Microscope operation and image pre-processing were performed in LAS-X (Leica Microsystems). Images were pixel binned by a factor of 5 (resulting in a final pixel size of 675 nm/px) to increase signal-to-noise and confidence in the photon arrival times. Thresholding was used to remove pixels only containing background photons (less than 25–50 counts depending on labelling). The lifetime images were generated using the Phasor-FLIM analysis pipeline integrated into LAS-X. Phasor-plots were median filtered (window size of 5) and a rainbow false colouring was applied from 3.75 to 4.75 ns. This was pure to aid visualisation of tension differences across the epiblasts. For quantitative analysis the Phasor-FLIM images were exported to.tiff (using 0.01 lifetime values per grey level to ensure accuracy). The .tiff-files contained the intensity images as well as the lifetime images and were further processed in Fiji[80] using a custom-written macro available at https://github.com/Faldalf/Royer_et_al_FLIM_ROIs.git. Briefly, the intensity image or the RGB phasor image was loaded to define regions of interest (ROIs, e.g. for the emVE or exVE), the ROIs were then applied to the exported lifetime image and the lifetime values per pixel within each region saved to a.csv file. The lifetime values from multiple embryos (nine $ASPP2^{EpiWT/ΔE4}$ embryos as controls and seven $ASPP2^{EpiΔE4/ΔE4}$ embryos) were further processed in R to calculate the mean lifetimes at the apical surface of each tissue or the whole epiblast using the tension-sensitive lifetime range of 2.8–7 ns[43].

**Embryo culture in channels**. Channels were formed by casting a 5% (which corresponded to ~4.2 kPa stiffness[83]) acrylamide hydrogel (containing 39:1 bis-crylamide) around 60 μm wires within the confinement of a two-part mould (10 mm × 10 mm × 1 mm). Ammonium persulphate (0.1%) and TEMED (1%) were added to polymerise polyacrylamide. The wires were then removed to form cylindrical cavities within hydrogel pieces. The hydrogels were carefully washed and equilibrated in embryo culture media at 37 °C and 5% $CO_2$. The embryos were then inserted into the channels using a glass capillary with a diameter slightly larger than the embryo itself. It was used to stretch the hydrogel channel before injecting the embryos and letting the channels relax and deform the embryos. Cell viability in channels had previously been assessed without any noticeable difference with control embryos[46]. After 30 min, embryos were fixed inside the hydrogels with 4% PFA for 35 min. Once fixed, embryos were removed from the hydrogel channels and wholemount immunohistochemistry was performed.

**Co-immunoprecipitation and SDS-PAGE/Immunoblotting**. For immunoprecipitation experiments, Caco-2 cells from confluent 10 cm diameter dishes were washed twice with PBS and then lysed in 500 μl of a buffer containing 50 mM Tris-HCl at pH 8, 150 mM NaCl, 1 mM EDTA, Complete Protease Inhibitor Cocktail (Roche) and 1% Triton X-100. Lysates were left on ice for 30 min, briefly sonicated and spun down at $21,000 × g$ for 30 min at 4 °C. The supernatant was transferred to another tube and protein concentration was measured (Bradford, Bio-Rad). About 1 mg of protein lysate was used per condition. Lysates were precleared using 20 μl protein G Sepharose 4 fast flow (1:1 in PBS, GE Healthcare) for 30 min at 4 °C on a shaker. The supernatant was incubated for 30 min at 4 °C on a shaker with 2 μl of the indicated antibody. About 30 μl protein G Sepharose 4 Fast Flow (1:1 in PBS) was added to each condition and samples were incubated overnight at 4 °C on a shaker. Samples were washed five times with ice-cold lysis buffer. About 25 μl sample buffer was added and samples were incubated at 95 °C for 5 min before being subjected to SDS-PAGE/Immunoblotting.

**Mesoderm explants and mesoderm cell migration**. $ASPP2^{WT/RAKA}$ mice heterozygous for the LifeAct-GFP transgene were crossed and E7.5 embryos were dissected in M2. Embryos were then incubated in a 2.5% pancreatin mixture on ice for 20 min. Using tungsten needles, the visceral endoderm layer was removed and then the mesodermal wings were separated from the underlying epiblast. Mesodermal tissue was grown in fibronectin-coated eight-well Lab-Tek II chamber slides and cultured in DMEM containing 10% foetal bovine serum, penicillin and streptomycin at 37 °C and 5% $CO_2$[84]. Samples were imaged on a Zeiss LSM 880 confocal microscope equipped with an environmental chamber to maintain conditions of 37 °C and 5% $CO_2$. A laser excitation wavelength of 488 nm was used, and explants were imaged every 5 mins for 5 h. For each time point, nine z-section with 1 μm step were collected.

Individual cells migrating away from the explants were tracked using the manual tracking plugin in Fiji[80]. The movement, velocity and directionality of individual cells was analysed. The movement represented the total distance travelled in μm by an individual cell. Velocity represented the average speed in μm/min of a given cell. Directionality was used as a measure of how direct or

convoluted a cell's path was and was calculated as the ratio between the total distance travelled and the distance in a straight line between a cell's start and end position[85].

**Embryonic stem cell-derived cysts**. Using small-molecule inhibitors of Erk and Gsk3 signalling[86], ASPP2[flE4/flE4] and ASPP2[RAKA/RAKA] (and ASPP2[WT/WT] controls) ESC were generated from flushed E2.5 embryos obtained from crosses between ASPP2[flE4/flE4] and ASPP2[WT/RAKA] mice, respectively. Briefly, embryos were grown for two days in organ culture dishes, containing pre-equilibrated preimplantation embryo culture media supplemented with 1 μM PDO325901 and 3 μM CHIR99021 (Sigma-Aldrich). Embryos were grown one more day in NDiff 227 media (Takara) supplemented with 1 μM PDO325901 and 3 μM CHIR99021 (NDiff + 2i). The trophectoderm was removed by immunosurgery and 'epiblasts' were grown in gelatinised dishes in the presence of NDiff + 2i and ESGRO (recombinant mouse LIF Protein, Millipore) to establish ESC lines.

ASPP2[flE4/flE4] ESC were infected with an Ad-CMV-iCre adenovirus (Vector Biolabs) to delete exon 4 of ASPP2. Deletion of exon 4 was assessed by PCR. Non-infected ASPP2[flE4/flE4] ESC were used as controls. Wild type ESC derived from littermates were used as controls for ASPP2[RAKA/RAKA] ESC. To form cysts, 4500 ESC were resuspended in 150 μl Matrigel (354230, Corning) and plated into a well of an eight-well Lab-Tek II chamber slide. The gel was left to set for 10 min at 37 °C before 300 μl differentiation medium (DMEM supplemented with 15% FCS, 1% Penicillin/Streptomycin, 1% Glutamine, 1% MEM non-essential amino acids, 0.1 mM 2-mercaptoethanol and 1 mM sodium pyruvate) was added. ESC were grown for 72 h at 37 °C and 5% $CO_2$ before immunostaining was performed.

**Statistics and reproducibility**. No statistical method was used to predetermine sample size. Embryos damaged during dissection were excluded from any analyses. The experiments were not randomised. The Investigators were not blinded to allocation during experiments and outcome assessment. Homozygous mutant embryos were compared to representative images of either wild type or heterozygous embryos as those were phenotypically undistinguishable.

**Reporting summary**. Further information on research design is available in the Nature Research Reporting Summary linked to this article.

## Data availability

The complete data supporting the results presented in this study are available upon a reasonable request from the corresponding authors. Source data are provided with this paper.

## Code availability

The custom-written macro used to analyse the Phasor-FLIM images in Fiji is available at https://github.com/Faldalf/Royer_et_al_FLIM_ROIs.git.

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

## Acknowledgements

This work was funded by Wellcome Senior Investigator Award 103788/Z/14/Z (S.S.). X.L., E.S. and F.Z. were funded by the Ludwig Institute for Cancer Research. F.S. was supported through funding from Wellcome (212343/Z/18/Z) and EMBO (ALTF 849-2020). B.C.L. was supported by MRC (MC_UU_12009 and MC_UU_12010). The generation and initial establishment of the *ASPP2*^WT/RAKA mice used in this study was funded by the Ludwig Institute for Cancer Research. The Leica SP8 Falcon was acquired with funds from WT (WT104924A1A) and the Wellcome Trust Institutional Strategic Support Fund (WT 0009773). We thank Jenny Nichols for advice and protocols for deriving Embryonic Stem Cells.

## Author contributions

C.R. and S.S. led the project, conceived and designed the experiments and analysed the data. C.R. performed the experiments unless otherwise stated. E.Sa. established the following mouse lines: *ASPP2*^flE4/flE4, *ASPP2*^WT/ΔE4, *ASPP2*^WT/ΔE4 mice homozygous for the *Sox2Cre* transgene, *ASPP2*^WT/ΔE4 mice homozygous for the mT/mG transgene. E.Sa. also performed the somite analysis and the sarcomeric α-actinin immunostaining. E.Sl. and X.L. designed and established the ASPP2^WT/RAKA mouse line. B.C.L. and F.S. performed and analysed FLIM experiments. M.F. coordinated FLIM experiments. J.Go. performed siRNA microinjections. K.L. inserted embryos into channels. N.V. performed immunostaining in MDCKII cells. J.Ga. analysed ESC cyst data. T.N. analysed mesoderm explants data. H.H. and F.Z. performed the unwrapping of ASPP2 and F-actin immunostaining in E5.5 embryos. A.V., C.J., T.C., K.C. and C.G. organised the collection of human embryos. C.R. performed the statistical analyses. C.R. and S.S. wrote the manuscript.

## Competing interests

The authors declare no competing interests.
