## [Peer Review File · Nature Communications]

ASPP2 maintains the integrity of mechanically stressed pseudostratified epithelia during morphogenesisReviewers' Comments:

Reviewer #1:

Remarks to the Author:

This manuscript describes the role of the ASPP2, a component of apical junctions, in the maintenance of the architecture of pseudostratified epithelia during mouse embryogenesis. The authors focus on the epiblast and to a lesser extent on pseudostratified epithelia in the somites and neural tube. Loss of function and imaging experiments have led to the conclusion that ASPP2 maintains epithelial architecture by preventing cells from escaping into the luminal space during mitosis. This is an interesting discovery that has multiple implications for development studies and application to other important fields of research such as cancer research.

This is a carefully conducted study. The authors have used multiple markers to analyze the phenotype of ASPP2 mutants, generated two different alleles of the gene and analyzed the phenotype in two different genetic backgrounds. They provide convincing evidence that absence of ASPP2 leads to structural failure of pseudostratified epithelia through its action on F-actin. They also show that ASPP2 interacts with F-actin via Afadin and provide convincing evidence that accumulation of ASPP2 cells in the proamniotic cavity of gastrulating embryos is due to their inability to sustain increased tension in the epiblast epithelium.

One of the main assertions in the study is that ASPP2 controls the formation of the proamniotic cavity. This is erroneous since the proamniotic cavity is visible in several mutant embryos (see Figure 1). A more accurate description, acknowledged by the authors, is that the proamniotic cavity is overwhelmed by an excess number of cells derived from the epiblast, obscuring its view.

Previous studies have shown that cells devoid of contact with the basal membrane in double-embryo chimeras undergo apoptosis and accumulate in the proamniotic cavity (Orietti et al., Stem Cells Reports, 2020). Aurora A Kinase epiblast mutant cells also undergo apoptosis and accumulate in the proamniotic cavity (Yoon et al., Dev. Biol., 2012). ASPP2 mutant epiblast cells located in the proamniotic cavity appear to undergo cell division and show no evidence of apoptosis, however, this possibility has not been explored by the authors.

The phenotype of ASPP2 mutants is prominent only in epiblast cells or in other pseudostratified epithelia such as somites and neural tube progenitors. Why is the absence of ASPP2 not affecting other tissues like visceral endoderm where the gene is also expressed? Contrary to the epiblast, the apical surface of visceral endoderm cells is not under mechanical stress to form a cavity. Also, one peculiarity of the epiblast, noted by the authors, is that ASPP2 is located at tricellular junctions but this is not the case in the visceral endoderm epithelium. What are the implications of these observations? It would be interesting if the authors could speculate why different epithelia behave differently in mutant embryos.

There is no mention of the phenotype of cells in the extra-embryonic ectoderm a tissue that also forms an epithelium facing the lumen of the proamniotic cavity. Do these cells present the same phenotype as epiblast cells? Do they show accumulation of ASPP2 at tricellular junctions?

Reviewer #2:

Remarks to the Author:

Maintaining epithelial architecture and cell polarity are critical for normal embryonic development and homeostasis and are often disrupted during oncogenesis. Here the authors present exciting new data analyzing the role of the protein ASPP2 in early mouse development, revealing striking roles in cell reintegration after cell division and in maintaining epithelial architecture. They combine stunning imaging of mouse embryos with careful quantitative analysis to dig deeply into the cell biological basis

of the embryonic phenotypes. I found most of their conclusions well supported and the implications will be of broad interest. As I outline below, there are a few conclusions that are not as well supported and could be softened, there is one simple experiment that seemed missing, and they could expand their Introduction to better provide context for the current work. All of these issues should be straightforward to address.

Introduction. The Introduction was surprisingly brief and could have been better focused. It includes a long paragraph about interkinetic nuclear migration which is not clearly relevant, and lacks any substantive information about what we already know about ASPP and its family, both biochemically from studies in cultured cells, or about its known roles in the mouse and in *Drosophila*.

Early phenotype-Figure 2. The early phenotype is quite striking, with massive accumulation of cells inside the epiblast. However, it was less clear how one proceeded from no phenotype at day 5.5 to this drastic phenotype at day 6.5. The data in Figure 3 was helpful in this regard but I was left wondering if more images from the intervening stage might be helpful. One obvious missing piece, in my mind, is whether loss of ASPP2 affects localization of E-cadherin. Are the cells in the epiblast unpolarized but adherent or have they lost cell adhesion?

Figure 3. The data in this Figure was stunning! The authors should look at data about re-integration in the mouse skin from (Elife. 2019 Dec 13;8:e49249) and in the *Drosophila* follicle cell epithelium from Dan Bergstralh. They provide interesting comparisons. The former paper might make them put more weight on the basal foot hypothesis.

Figure 4. The cortical actin phenotype is quite interesting, as it seems reasonably variable. Some cells (Fig 4C arrow) seem to have lost epithelial architecture while others (above the arrow) are more normal. This deserves more mention. The elevated protrusive behavior of cells Figure 4EF was also stunning and deserved more mention. Is this reflective of a partial EMT?

Figure 5. I loved the “confinement experiment”! I would note that I would interpret the images in Fig 5A differently. I do not think myosin is activated in the rounded up dividing cells but instead is elevated in their neighbors, who have to adjust their contractility in response. This is exactly what is seen in *Drosophila* embryos in cells near the dividing mitotic domains (e.g. Mol Biol Cell. 2019 Jul 22;30(16):1938-1960). In Figure 5C they note “Interestingly, F-actin and Myosin were abnormally distributed at the apical surface of cells accumulating ectopically in ASPP2RAKA/RAKA mutant embryos, suggesting that actomyosin contractility was disrupted.” In contrast, to me the cells they are highlighting look like they are apically constricting, and they are right at the most curved region of the epiblast.

Fig. 6. I thought the parallel to Afadin was very interesting, though the TCJ enrichment was overstated and should be toned down. There is a lot more to discuss about this, using the fly as an example—ASPP and Canoe have very interesting parallel phenotypes in cell shapes in the developing pupal eye—e.g., see Dev Cell. 2019 Aug 5;50(3):313-326 and earlier work on ASPP. This Figure also had the only data I found unconvincing. The authors state: During cell division events, following mitotic rounding, apical F-actin localization was disrupted in ASPP2RAKA/RAKA embryos whereas it was maintained in wild type embryos (Fig. 6g).” This was not convincing

The Discussion could be refocused on the comparisons to earlier work in *Drosophila* and the mouse, and to address the issues above. The focus on PP1 regulation, mentioned prominently in the Abstract, is never really explored directly and should be toned down. Do we know if their RAKA mutant even makes a stable protein?

Mark Peifer

Reviewer #3:

Remarks to the Author:

Apoptosis-stimulating protein of p53 (ASPP2 or Tumor protein p53 binding protein 2 [TP53BP2]) was reported to be essential to epithelial cell polarity via Par-3 and apoptosis/cell growth through the p53 family. In this manuscript, the authors analyze the epithelial phenotypes of *Aspp2* Δ E4/ Δ E4 and *Aspp2* RAKA/RAKA mutant embryos in terms of cell polarity. New findings are related to the interaction of PP1 with ASPP2 by analyzing *Aspp2* RAKA/RAKA mutant embryos in Figures 6 and 7. They finally conclude that *Aspp2* is an essential component of a system that maintains tissue integrity under conditions of increased mechanical stress in a broad range of tissues. Although these findings are potentially interesting, their group has reported *Aspp2* mutant phenotypes previously (Dev Cell 2010, PLoS one 2014). Based on several *Aspp2* papers published including the above, novel points of the current study are not demonstrated well as follows.

Major concerns:

1) Redundant roles of ASPP proteins

Since it is well known that ASPP1 is highly similar to ASPP2 in structure and function (Bergamaschi et al., 2004; Samuels-Lev et al., 2001), ASPP1 may be able to rescue ASPP2 throughout development including control of the Hippo pathway in trophoblast and epithelial polarity in visceral endoderm cells. Therefore, it would be very important to analyze *Aspp2/Aspp1* double KO embryos during the pre- and post-implantation stages.

Related to this issue, the authors use two different genetic backgrounds, C57BL/6J and BALB/c. These phenotypic varieties might be due to the redundant roles played by two *Aspp* genes, they could use specific Cre driver lines to clarify the roles of *Aspp2/Aspp1* on your interest.

ASPP1 and ASPP2: common activators of p53 family members.

Bergamaschi D, Samuels Y, Jin B, Duraisingham S, Crook T, Lu X. Mol Cell Biol. 2004

ASPP proteins specifically stimulate the apoptotic function of p53.

Samuels-Lev Y, O'Connor DJ, Bergamaschi D, Trigiante G, Hsieh JK, Zhong S, Campargue I, Naumovski L, Crook T, Lu X. Mol Cell. 2001

2) Genetic interaction between Pp1 and *Aspp2*

The authors exploit the RAKA mutant allele of the *Aspp2* gene, which results in no association with PP1. Finally, as in their title, they conclude that phenotypes of *Aspp2* RAKA/RAKA embryos reflect the PP1 function. However, this hypothesis is not proved yet but may be incorrect. To verify this, they should also evaluate whether the phenotypes of *Aspp2* RAKA/RAKA embryos they concentrated on are genetically interacted with Pp1-deficiency by crossing Pp1 mutant mice with *Aspp2* RAKA/+ and *Aspp2* Δ E4/+ mutant mice.

3) Quantitative value of mechanical stress

There is no biophysical evidence demonstrating that defected tissues and regions show stiffer mechanical stress than other unaffected regions in *Aspp2* mutant embryos. If their hypothesis was correct, cell cortex tension of *Aspp2* mutant tissues and cells would be softer or more fragile than those of the wild type. You should measure the stiffness of the affected tissues quantitatively with an atomic force microscope directly or with a tension sensor imaging system indirectly. Analysis with p-Myosin expression is not quantitative.

4) Primary defects of *Aspp2* mutant embryos.

It remains unclear if mutant defects are primarily due to failure in epithelial polarity, cell growth/apoptosis, or mechanical stress. How epithelial polarity is linked to amniotic cavity formation? They should clarify whether the phenotypes of *Aspp2* mutant embryos in Figure 5 are caused by the overgrowth/apoptosis or mechanical failure.

Besides, the authors might misinterpret normal epithelial-mesenchymal transition (EMT) and overgrowth of mesodermal cells as defects in epithelial polarity (in Figure 4). To clarify two phenomena of overgrowth of mesodermal cells and polarity failure explicitly, they could analyze specific markers expression such as EMT- and germ layer-specific markers including E-cadherin, N-cadherin, Tbr1, and so on.

5) Phenotypes of mutant visceral endoderm

Except for epiblast defects, extraembryonic specific markers should be examined for mutant embryos, in particular, for the visceral endoderm (VE) layer; AVE/ DVE, pan-VE markers, and so on.

Additionally, since *Aspp2* is expressed in the VE layer (in Figure 6), epithelial polarity defects can be observed in the mutant VE cells. Moreover, they could identify the epithelial polarity defects more specifically than epiblast cells because mutant VE cells appear to show little overgrowth as compared to epiblast cells do.

If you could not find polarity defects in the *Aspp2* mutant VE layer, *Aspp1* may complement the function of the *Aspp2* gene in VE cells.

RESPONSE TO REVIEWER COMMENTS

We thank the reviewers for their constructive criticism, in response to which we have now modified our manuscript with results from the following additional experiments:

Bullet point list of all the additional data

- Extended Data Fig 2c: 3D opacity rendering showing the localisation pattern and expression of GATA6, F-actin and Par6 in the outside VE layer of E6.5 wild type and *ASPP2*^{ΔE4/ΔE4} embryos.
- Extended Data Fig. 2d: Images showing the normal apical localisation of Par6 and F-actin in the EXE region of wild type and *ASPP2*^{ΔE4/ΔE4} embryos.
- Extended Data Fig. 2e: Immunostaining showing that the localisation of E-cadherin is only disrupted near the apical junctions in epiblast cells of *ASPP2*^{EpiΔE4/ΔE4} embryos.
- Extended Data Fig. 2f: Analysis of the proportion of PHH3-positive cells in control and *ASPP2*^{EpiΔE4/ΔE4} embryos showing no significant difference.
- Extended Data Fig. 2g: Immunostaining showing SOX17 expression in the definitive endoderm of control and *ASPP2*^{EpiΔE4/ΔE4} embryos.
- Extended Data Fig. 2h: Cleaved Caspase-3 Immunostaining in control and *ASPP2*^{EpiΔE4/ΔE4} embryos.
- Extended Data Fig. 4b: *ASPP2* immunostaining showing comparable expression at the apical junctions in the VE between wild type and *ASPP2*^{RAKA/RAKA} embryos.
- Fig. 5a,b and Extended Data Fig. 5: Analysis of apical mechanical stress in various tissues of E6.5 embryos using FLIPPER-TR membrane tension-sensitive probe in conjunction with fluorescence lifetime imaging microscopy (FLIM).
- Fig. 5c: Analysis of SHROOM2 localisation in wild type E6.5 embryos, showing its specific enrichment at the apical junctions of the epiblast.
- Fig. 6i and Extended Data Fig. 6g,h, i: Analysis of tensions in wild type and *ASPP2*^{EpiΔE4/ΔE4} E6.5 embryos using FLIPPER-TR membrane tension-sensitive probe in conjunction with fluorescence lifetime imaging microscopy (FLIM).

We provide a point-by-point response to each of the comments below.

Reviewer #1 (Remarks to the Author):

This manuscript describes the role of the ASPP2, a component of apical junctions, in the maintenance of the architecture of pseudostratified epithelia during mouse embryogenesis. The authors focus on the epiblast and to a lesser extent on pseudostratified epithelia in the somites and neural tube. Loss of function and imaging experiments have led to the conclusion that ASPP2 maintains epithelial architecture by preventing cells from escaping into the luminal space during mitosis. This is an interesting discovery that has multiple implications for development studies and application to other important fields of research such as cancer research.

This is a carefully conducted study. The authors have used multiple markers to analyze the phenotype of ASPP2 mutants, generated two different alleles of the gene and analyzed the phenotype in two different genetic backgrounds. They provide convincing evidence that absence of ASPP2 leads to structural failure of pseudostratified epithelia through its action on F-actin. They also show that ASPP2 interacts with F-actin via Afadin and provide convincing evidence that accumulation of ASPP2 cells in the proamniotic cavity of gastrulating embryos is due to their inability to sustain increased tension in the epiblast epithelium.

We would like to thank this reviewer for their overwhelmingly positive comments and constructive criticism.

One of the main assertions in the study is that ASPP2 controls the formation of the proamniotic cavity. This is erroneous since the proamniotic cavity is visible in several mutant embryos (see Figure 1). A more accurate description, acknowledged by the authors, is that the proamniotic cavity is overwhelmed by an excess number of cells derived from the epiblast, obscuring its view.

We agree. Our results show that cells in the epiblast of E5.5 $ASPP2^{\Delta E4/\Delta E4}$ embryos are initially able to polarise and form a cavity (Extended Data Fig. 2a). As apical daughter cells escape the epiblast, they progressively occlude the cavity (Figure 3). We have therefore made changes to the text throughout the manuscript to accurately reflect that and underline the fact that ASPP2 is required to maintain the architecture of the proamniotic cavity rather than control its formation.

Previous studies have shown that cells devoid of contact with the basal membrane in double-embryo chimeras undergo apoptosis and accumulate in the proamniotic cavity (Orietti et al., Stem Cells Reports, 2020). Aurora A Kinase epiblast mutant cells also undergo apoptosis and accumulate in the proamniotic cavity (Yoon et al., Dev. Biol., 2012). ASPP2 mutant epiblast cells located in the proamniotic cavity appear to undergo cell division and show no evidence of apoptosis, however, this possibility has not been explored by the authors.

To address this point, we performed phospho-histone H3 and cleaved Caspase-3 Immunostainings in control and $ASPP2^{Epi\Delta E4/\Delta E4}$ embryos. At E6.5, we could not find significant differences in the proliferation rate of epiblast cells between control and $ASPP2^{Epi\Delta E4/\Delta E4}$ embryos (Extended Data Fig. 2f). We therefore rule out the possibility that cells escaped into the proamniotic cavity because of an excess of proliferation in the epiblast.

Additionally, very few epiblast cells were cleaved Caspase-3-positive in either control or *ASPP2*^{EpiΔE4/ΔE4} embryos at E6.5 (Extended Data Fig. 2h). We could only observe clear differences between control and *ASPP2*^{EpiΔE4/ΔE4} embryos at E7.5, when cells of *ASPP2*^{EpiΔE4/ΔE4} embryos had already accumulated to such an extent in the centre that they had presumably run out of room and started dying (Extended Data Fig. 2g). We therefore think that the phenotype we observe is different to the one observed in Aurora A epiblast-null embryos (Yoon et al., *Dev. Biol.*, 2012) or the result of a growth control mechanism such as that which occurs in double embryo chimeras (Orietti et al., *Stem Cells Reports*, 2020).

The phenotype of *ASPP2* mutants is prominent only in epiblast cells or in other pseudostratified epithelia such as somites and neural tube progenitors. Why is the absence of *ASPP2* not affecting other tissues like visceral endoderm where the gene is also expressed? Contrary to the epiblast, the apical surface of visceral endoderm cells is not under mechanical stress to form a cavity. Also, one peculiarity of the epiblast, noted by the authors, is that *ASPP2* is located at tricellular junctions but this is not the case in the visceral endoderm epithelium. What are the implications of these observations? It would be interesting if the authors could speculate why different epithelia behave differently in mutant embryos.

We have modified the discussion to consider why different epithelia that express *ASPP2* might show different sensitivities to the loss of its function. There are many potential explanations for this to be the case, such as for example, different molecules working in conjunction with *ASPP2* in the epiblast that are not present in other tissues. Our new results highlight crucial differences between the epiblast and other epithelial tissues at this particular stage in development.

We have now also performed additional experiments to estimate tissue tensions at E6.5 using a FLIPPER-TR probe and fluorescence lifetime imaging. We found that lifetimes were highest at the apical surface of the epiblast around the proamniotic cavity in comparison to the apical surface of the embryonic VE or extraembryonic VE, suggesting that tensions are highest there (Fig. 5a,b and below). We also provide evidence that SHROOM2, a protein that can interact with F-actin, Myosin and ZO-1, is enriched at the apical junctions, specifically in the epiblast (Fig. 5c and below). Considering the importance of SHROOM proteins in the regulation of apical actomyosin contractility, the higher apical tensions observed in the epiblast may be due to cells apically constricting. This would agree with the phenotype observed in the primitive streak of *ASPP2*^{RAKA/RAKA} embryos where cells accumulate in the region of the primitive streak where apical constrictions also play an important role. This new set of evidence also agrees with our data showing that increasing mechanical stress can cause earlier phenotypes in the epiblast of *ASPP2*^{RAKA/RAKA} embryos. Together, this leads us to believe that the apical junctions in the epiblast are under higher mechanical stress which could explain why *ASPP2* is specifically required in this context. The enriched localisation of *ASPP2* at tri- or multi-cellular junctions in the epiblast is consistent with this tissue experiencing increased apical tensions. We also think that the overall configuration of pseudostratified epithelia - high cellularity and narrow apical domain - is likely to contribute to the impact of cell rounding during mitosis on tensions exerted at the level of apical junctions.

There is no mention of the phenotype of cells in the extra-embryonic ectoderm a tissue that also forms an epithelium facing the lumen of the proamniotic cavity. Do these cells present the same phenotype as epiblast cells? Do they show accumulation of *ASPP2* at tricellular junctions?

We now include new data showing the absence of an obvious phenotype in the extra-embryonic ectoderm (Extended Data Fig. 2d and below), including Par6 stains showing normal polarisation of these cells. We include in the revised manuscript a discussion of why it might be that the ExE, similar to the VE, does not show a phenotype in mutants.

Reviewer #2 (Remarks to the Author):

Maintaining epithelial architecture and cell polarity are critical for normal embryonic development and homeostasis and are often disrupted during oncogenesis. Here the authors present exciting new data analyzing the role of the protein ASPP2 in early mouse development, revealing striking roles in cell reintegration after cell division and in maintaining epithelial architecture. They combine stunning imaging of mouse embryos with careful quantitative analysis to dig deeply into the cell biological basis of the embryonic phenotypes. I found most of their conclusions well supported and the implications will be of broad interest. As I outline below, there are a few conclusions that are not as well supported and could be softened, there is one simple experiment that seemed missing, and they could expand their Introduction to better provide context for the current work. All of these issues should be straightforward to address.

We would like to thank you for your very positive and thoughtful comments and constructive criticism of our manuscript Mark.

Introduction. The Introduction was surprisingly brief and could have been better focused. It includes a long paragraph about interkinetic nuclear migration which is not clearly relevant and lacks any substantive information about what we already know about ASPP and its family, both biochemically from studies in cultured cells, or about its known roles in the mouse and in *Drosophila*.

In line with what you are suggesting, we have added a paragraph to the introduction to add concise background information on ASPP2 relevant to this manuscript (Line 75 to 85).

We thought it was important to also introduce interkinetic nuclear migration as it is a fundamental characteristic of pseudostratified epithelia (where ASPP2 plays a fundamental role). We think that a description of interkinetic nuclear migration helps set the stage and explain why pseudostratified epithelia in general may be under more mechanical stress. For example, one can imagine that cells rounding up prior to cell divisions at the apical surface might exacerbate mechanical stress at the apical junctions of an already crowded epithelium.

Early phenotype-Figure 2. The early phenotype is quite striking, with massive accumulation of cells inside the epiblast. However, it was less clear how one proceeded from no phenotype at day 5.5 to this drastic phenotype at day 6.5. The data in Figure 3 was helpful in this regard but I was left wondering if more images from the intervening stage might be helpful.

It has been difficult to obtain fixed samples that capture the very onset of the phenotype for two reasons. Firstly, in a given litter, the penetrance of this phenotype varies considerably (Fig. 2a). Secondly, as our live imaging data suggests (Fig. 6h and Extended Data Fig. 6f), from the moment the defect becomes apparent, the space of the proamniotic cavity adjacent to the epiblast becomes extremely rapidly filled with cells (1-2 hours approximately). In an effort to represent the progression of the defect, we have attempted to show embryos with different phenotype severities throughout the manuscript. It was not always possible to show early defects for all immunostainings considering how hard it is to capture; we show several examples in the manuscript nonetheless (Fig. 2h for example).

One obvious missing piece, in my mind, is whether loss of ASPP2 affects localization of E-cadherin. Are the cells in the epiblast unpolarized but adherent or have they lost cell adhesion?

We are grateful for your suggestion and have now included an E-cadherin immunostaining. Interestingly, we found something very similar to what we saw with SCRIB. In wild type epiblasts, just like SCRIB, E-cadherin was found basolaterally but was also enriched closer to the apical domain (examples of this can be found throughout the literature on mouse preimplantation development, please see this paper from Bedzhov et al., Cell, 2014, DOI: 10.1016/j.cell.2014.01.023 for example). Our new data indicates that, in *ASPP2*^{Epi Δ E4/ Δ E4} E6.5 embryos, adhesion was not completely lost in epiblast cells (Extended Data Fig. 2e). However, the enrichment of E-cadherin at the apical junctions was partially lost and could not be seen in the cells being abnormally extruded into the proamniotic cavity. Again, this suggests that failure of the apical junctions rather than lateral adhesion is responsible for the defect.

Figure 3. The data in this Figure was stunning! The authors should look at data about re-integration in the mouse skin from (Elife. 2019 Dec 13;8:e49249) and in the *Drosophila* follicle cell epithelium from Dan Bergstralh. They provide interesting comparisons. The former paper might make them put more weight on the basal foot hypothesis.

We are again very grateful for this excellent suggestion. We have added these elements to the discussion as noted below. In regard to the basal endfoot hypothesis, this is something we have tried to look into. However, as all cell membranes were labelled in our live imaging experiments and as the epiblast is extremely crowded, we were not able to clearly discern these potential basal endfeet. Instead, this is something that should be tested in mosaically-labelled epiblasts in future experiments. To reflect this, we added the following discussion elements:

Line 424 to 437:

“However, it remains unclear why it is only the more apically localised daughter cells that are affected in ASPP2 mutant embryos. One possibility is that these daughters do not inherit the basal process that tethers cells to the basement membrane in pseudostratified epithelia and rely on intact apical F-actin organisation and tensions to reintegrate basally. However, the idea that the basal process is inherited asymmetrically is controversial⁵². Interestingly, a study in the epidermis suggests that some apical daughters retain a basal endfoot that enables the reorientation of cell divisions within the plane of the epithelium. Moreover, this study shows that Afadin functions in endfoot retention⁵³. We could not clearly establish whether daughter cells were making contact via a basal process in our live imaging experiments mainly because of the high cellular density of the epiblast. However, considering the link between ASPP2 and Afadin, it will be interesting to test this hypothesis further using, for example, mosaically-labelled epiblast cells. Studies in *Drosophila* have suggested there exist alternative mechanisms involving lateral adhesion mediated by immunoglobulin superfamily cell adhesion molecules that support daughter cell reintegration^{54,55}. Our data suggests that the apical junctions

may play an equivalent role in the epiblast and more generally in pseudostratified epithelia.”

Figure 4. The cortical actin phenotype is quite interesting, as it seems reasonably variable. Some cells (Fig 4C arrow) seem to have lost epithelial architecture while others (above the arrow) are more normal. This deserves more mention.

We have now added additional labelling and magnifications to attempt to clarify Figure 4c (please see below). At E7.5, *ASPP2*^{RAKA/RAKA} embryos exhibit disrupted epiblast architecture exclusively in the primitive streak region, in the posterior of the embryo (now outlined in orange dotted line). In the more lateral regions of the epiblast, tissue architecture remains intact (now outlined in green dotted line). This is perhaps more obvious in Figure 4f, when looking directly at the posterior of the embryo from within the proamniotic cavity. In the central part of the image corresponding to the primitive streak (between the green dotted lines), F-actin organisation is disrupted, whereas the more lateral regions of the epiblast show normal F-actin organisation (outside the green dotted lines).

To reflect changes to Figure 4c, we have also made adjustments to the text as below.

Line 245 to 249:

“In contrast, *ASPP2*^{RAKA/RAKA} embryos exhibited a clear ectopic accumulation of cells apical to the primitive streak as visualised by T expression (Fig. 4b). In these cells, F-actin was abnormally uniformly distributed along the apical surface, with no clear apical-junction enrichment. This was specific to cells in the posterior of the epiblast where the primitive streak forms, as F-actin organisation was undisturbed in the lateral regions of the epiblast (Fig. 4c).”

The elevated protrusive behavior of cells Figure 4EF was also stunning and deserved more mention. Is this reflective of a partial EMT?

Our preliminary scRNAseq data suggests that the expression levels of EMT-related mRNA are unchanged between wild type and *ASPP2*^{RAKA/RAKA} cells in a BALB/c background in the epiblast and the primitive streak. In fact, we see very little transcriptional changes overall, in line with our stains for tissue specific markers (Fig. 4b), which show the correct specification of cell types, but profound morphological defects. None of the differentially expressed genes between wild type and

ASPP2^{RAKA/RAKA} embryos related to EMT. Below, we show several examples of EMT and epithelial markers that remained unchanged between wild type and *ASPP2*^{RAKA/RAKA} cells in the epiblast and the primitive streak.

Figure 5. I loved the “confinement experiment”! I would note that I would interpret the images in Fig 5A differently. I do not think myosin is activated in the rounded up dividing cells but instead is elevated in their neighbors, who have to adjust their contractility in response. This is exactly what is seen in *Drosophila* embryos in cells near the dividing mitotic domains (e.g. Mol Biol Cell. 2019 Jul 22;30(16):1938-1960).

We agree that our data does not allow us to establish with precision whether the cells expressing increased p-Myosin apically were cells rounding up or cells directly next to them as their apical domains are extremely close. In light of your comment and other reviewers’ comments, we have therefore decided to remove this figure panel. Instead, we are refocusing figure 5 on the differences between the epiblast and other tissues at E6.5, showing that the epiblast is under higher apical tension and that SHROOM2 is enriched at apical junctions in the epiblast.

In Figure 5C they note “Interestingly, F-actin and Myosin were abnormally distributed at the apical surface of cells accumulating ectopically in *ASPP2*^{RAKA/RAKA} mutant embryos, suggesting that actomyosin contractility was disrupted.” In contrast, to me the cells they are highlighting look like they are apically constricting, and they are right at the most curved region of the epiblast.

Cells in this curved region of the epiblast have clearly narrower apical domains suggesting that they are indeed apically constricting. However, this is true for both wild type and *ASPP2*^{RAKA/RAKA} embryos.

Our point is that cells of wild type embryos do so whilst conserving very distinct F-actin and Myosin enrichment at the apical junctions while this is not the case in *ASPP2*^{RAKA/RAKA} embryos.

However, we take the point that the conclusions from this observation are subject to interpretation, and we have therefore altered the text as follows.

Line 285 to 287:

“Interestingly, the localisation pattern of F-actin and Myosin at the apical surface of cells accumulating ectopically in *ASPP2*^{RAKA/RAKA} mutant embryos was altered in a way similar to that observed at E7.5 in cells accumulating at the surface of the primitive streak.”

Fig. 6. I thought the parallel to Afadin was very interesting, though the TCJ enrichment was overstated and should be toned down. There is a lot more to discuss about this, using the fly as an example—ASPP and Canoe have very interesting parallel phenotypes in cell shapes in the developing pupal eye—e.g., see *Dev Cell*. 2019 Aug 5;50(3):313-326 and earlier work on ASPP.

To take on board these comments, we made the following changes to the text:

Line 300 to 302:

“This revealed that, although *ASPP2* was uniform in its distribution along all junctions in the VE, in the epiblast, it was enriched in specific locations, often coinciding with F-actin-rich tricellular junctions.”

Line 454

Our results highlight the previously underappreciated ~~localisation pattern of *ASPP2* at tricellular junctions~~ discrete localisation pattern of *ASPP2* along the apical junctions in epithelial cells, in particular of the epiblast

Line 472

~~We therefore suggest that~~ It will therefore be important to test whether *ASPP2* ~~may be~~ is directly involved in the response to tissue tension by interacting with Afadin at the level of tricellular junctions to maintain F-actin organisation.

This Figure also had the only data I found unconvincing. The authors state: During cell division events, following mitotic rounding, apical F-actin localization was disrupted in *ASPP2*^{RAKA/RAKA} embryos whereas it was maintained in wild type embryos (Fig. 6g).” This was not convincing.

It was technically very challenging to image these embryos for long periods of time due to the relatively low signal intensity of the LifeAct-GFP transgene in the epiblast at these stages and the depth at which the imaging had to be done to capture the proamniotic cavity. We therefore had to compromise on image quality to reduce phototoxicity. However, from these movies we were able to see that the organisation of apical F-actin was disrupted locally prior to the accumulation of cells in the proamniotic cavity. Once this event took place, the proamniotic cavity filled up quickly. We have modified the text (please see below) and changed the title of this section to de-emphasise the relative importance given to these results, as we already show extensive characterisation of F-actin organisation in *ASPP2*^{RAKA/RAKA} mutants in a BALB/c background. We now refocus this section on the

potential role of ASPP2 in organising apical actomyosin contractility in the epiblast by providing evidence that tensions are overall reduced and disorganised in the absence of ASPP2.

Line 322:

“Together, these results highlight the importance of the localisation pattern of ASPP2 in the epiblast, suggesting that it may be able to interact with F-actin at the apical junctions via its interaction with Afadin. To further investigate the role of ASPP2 in maintaining the organisation of the F-actin cytoskeleton in the epiblast, we generated *ASPP2*^{RAKA/RAKA} embryos in a C57BL/6 background carrying a LifeAct-GFP transgene⁵¹. This allowed us to visualise F-actin in living embryos with time-lapse confocal microscopy (Fig. 6g,h and Extended Data Fig. 6f). In wild type embryos, we found that as epiblast cells divided, apical F-actin localisation was maintained (Fig. 6g). In contrast, in *ASPP2*^{RAKA/RAKA} embryos, apical F-actin organisation was locally disrupted (Fig. 6g) and this was followed rapidly by the abnormal extrusion of cells into the proamniotic cavity (Fig. 6h). These results suggest that ASPP2 function is required to maintain the architecture of apical F-actin in the epiblast and in its absence, actomyosin contractility at the apical junctions may be disrupted.”

The Discussion could be refocused on the comparisons to earlier work in *Drosophila* and the mouse, and to address the issues above.

We have modified our manuscript to now discuss the similarities in ASPP2- and Afadin-related phenotypes in greater details.

The focus on PP1 regulation, mentioned prominently in the Abstract, is never really explored directly and should be toned down. Do we know if their RAKA mutant even makes a stable protein?

We have now added a figure panel (Extended Data Fig. 4b and below) to show that ASPP2^{RAKA} mutant proteins are expressed and localise to the apical junctions in the VE of *ASPP2*^{RAKA/RAKA} embryos. We also reemphasise that the mutations present in this allele were designed to mimic previously extensively characterised ASPP2 mutations that have shown to abrogate the interaction between human ASPP2 and PP1 (S. Llanos et al., JBC, 2011) without affecting the ability of ASPP2 to localise normally at tight junctions in epithelial cells (C. Royer et al., PLOS ONE, 2014).

We take the point that we do not directly show the effect of the mutation on the recruitment of PP1 *in vivo* in embryos and have therefore made text changes throughout the manuscript to reflect this.

We also added the following to the text in relation to the new data presented in Extended Data Fig. 4b.

Line 237:

“Importantly, none of these defects were as a result of ASPP2^{RAKA} mutant proteins being unstable or mislocalised as they could be observed at the apical junctions in embryos at similar levels to wild type ASPP2 (Extended Data Fig. 4b)”

Reviewer #3 (Remarks to the Author):

Apoptosis-stimulating protein of p53 (ASPP2 or Tumor protein p53 binding protein 2 [TP53BP2]) was reported to be essential to epithelial cell polarity via Par-3 and apoptosis/cell growth through the p53 family. In this manuscript, the authors analyze the epithelial phenotypes of *Aspp2* Δ E4/ Δ E4 and *Aspp2* RAKA/RAKA mutant embryos in terms of cell polarity. New findings are related to the interaction of PP1 with ASPP2 by analyzing *Aspp2* RAKA/RAKA mutant embryos in Figures 6 and 7. They finally conclude that *Aspp2* is an essential component of a system that maintains tissue integrity under conditions of increased mechanical stress in a broad range of tissues. Although these findings are potentially interesting, their group has reported *Aspp2* mutant phenotypes previously (Dev Cell 2010, PLoS one 2014). Based on several *Aspp2* papers published including the above, novel points of the current study are not demonstrated well as follows.

We would like to thank this reviewer for their time to review our manuscript. In reply to this reviewer’s comments regarding the novelty of our findings, we would like to reemphasise some of the important novel points made by our findings:

- We show the previously unknown role for ASPP2 in the maintenance of proamniotic cavity architecture.
- We show that ASPP2 controls proamniotic architecture via apical daughter cell reintegration following cell divisions.
- We demonstrate that ASPP2 has a generalised role across several pseudostratified epithelia during development.
- We show that ASPP2 is required for apical F-actin organisation.
- Our data reveals that at these stages of embryogenesis, the function of ASPP2 is fulfilled via its PP1 binding site.
- We now also provide additional evidence supporting the hypothesis that ASPP2 is specifically required in epithelia that experience increased mechanical stress (please see below for more details).

Major concerns:

1) Redundant roles of ASPP proteins Since it is well known that ASPP1 is highly similar to ASPP2 in structure and function (Bergamaschi et al., 2004; Samuels-Lev et al., 2001), ASPP1 may be able to

rescue ASPP2 throughout development including control of the Hippo pathway in trophectoderm and epithelial polarity in visceral endoderm cells. Therefore, it would be very important to analyze *Aspp2/Aspp1* double KO embryos during the pre-and post-implantation stages. Related to this issue, the authors use two different genetic backgrounds, C57BL/6J and BALB/c. These phenotypic varieties might be due to the redundant roles played by two *Aspp* genes, they could use specific Cre driver lines to clarify the roles of *Aspp2/Aspp1* on your interest.

ASPP1 and ASPP2: common activators of p53 family members. Bergamaschi D, Samuels Y, Jin B, Duraisingam S, Crook T, Lu X. *Mol Cell Biol.* 2004

ASPP proteins specifically stimulate the apoptotic function of p53. Samuels-Lev Y, O'Connor DJ, Bergamaschi D, Trigiante G, Hsieh JK, Zhong S, Campargue I, Naumovski L, Crook T, Lu X. *Mol Cell.* 2001

It is well established that ASPP2 interacts with Par3 and localises to the apical/tight junctions in epithelial cells where it is known to function (Cong et al., *Current Biology*, 2010; Sottocornola et al., *Dev. Cell*, 2010; Royer et al., *PLOS ONE*, 2014, Matsuzawa et al., *Communications Biology*, 2021). Proteomics studies also show that ASPP2, and not ASPP1 nor iASPP, interacts with proteins related to tight junctions (Zhang et al., *Oncotarget*, 2015; Hennigan et al., *Science Signaling*, 2019). When ASPP1 and ASPP2 were used as baits in parallel, common binding partners were identified. However, ASPP2 exclusively interacted with a group of proteins relating to tight junctions (Par3, MAGI2, Afadin, etc), as opposed to ASPP1 (Zhang et al., *Oncotarget*, 2015). When Merlin was used as a bait, ASPP2, but not ASPP1 or iASPP, was identified as a binding partner, together with numerous other junctional proteins (Afadin, TJP1, TJP2, Shroom4, Par3, MAGI1 and 2, etc) (Hennigan et al., *Science Signaling*, 2019).

Moreover, published datasets show that ASPP1 expression is extremely low in all tissues at E7.5 (Blanca Pijuan-Sala et al., *Nature*, 2019; <https://marionilab.cruk.cam.ac.uk/MouseGastrulation2018/>). This is in agreement with our preliminary scRNAseq data (see below) which shows low expression at E7.5 in wild type embryos, and that it remains so in *ASPP2*^{RAKA/RAKA} tissues, suggesting that there is no compensatory upregulation of ASPP1.

scRNAseq data showing that ASPP1 mRNA are overall found at extremely low level in E7.5 embryos and those levels are comparable between wild type and *ASPP2*^{RAKA/RAKA} embryos.

While it might be interesting to clarify why loss of ASPP2 does *not* result in a phenotype in some tissues in which it is expressed, we are of the view it is more important to clarify why it *does* result in a profound phenotype in some tissues, and therefore chose to prioritise that question for this study.

Rather than a compensatory role of ASPP1 in the VE or at pre-implantation stages, we are of the view that there are more likely explanations, such as for example, different molecules working in conjunction with ASPP2 in the epiblast that are not present in other tissues. Our new results highlight crucial differences between the epiblast and other epithelial tissues at the egg cylinder stage.

We show that ASPP2 is specifically required in the epiblast where apical tensions are higher (Fig. 5a,b). We also show that the molecular organisation of apical junctions in the epiblast differs from that of other tissues such as the exVE, the emVE and the EXE. For example, we show that SHROOM2 is enriched at the apical junctions specifically in the epiblast (Fig. 5c), suggesting that apical constrictions are higher in this tissue, which agrees with apical tensions being higher in this tissue at E6.5. These observations, together with the requirement of ASPP2 in the primitive streak of *ASPP2*^{RAKA/RAKA} embryos where apical constrictions also play an important role, suggest that ASPP2 is required when cells actively constrict apically and are subject to increased mechanical stress. We added these elements to the discussion.

2) Genetic interaction between Pp1 and Aspp2. The authors exploit the RAKA mutant allele of the *Aspp2* gene, which results in no association with PP1. Finally, as in their title, they conclude that phenotypes of *Aspp2* RAKA/RAKA embryos reflect the PP1 function. However, this hypothesis is not proved yet but may be incorrect. To verify this, they should also evaluate whether the phenotypes of *Aspp2* RAKA/RAKA embryos they concentrated on are genetically interacted with Pp1-deficiency by crossing Pp1 mutant mice with *Aspp2* RAKA/+ and *Aspp2* Δ E4/+ mutant mice.

In light with this comment, we have changed the title and throughout the manuscript rephrased the conclusions drawn from experiments using *ASPP2*^{RAKA/RAKA} mutants, to de-emphasise the interaction with PP1 and retain focus on ASPP2. We now report the role of “the PP1-binding site of ASPP2” instead of the role of “the ASPP2/PP1 interaction”. However, we would like to draw this reviewer’s attention to the fact that mutations present in this allele were designed to mimic previously extensively characterised ASPP2 mutations that have shown to abrogate the interaction between human ASPP2 and PP1 (S. Llanos et al., JBC, 2011) without affecting the ability of ASPP2 to localise normally at tight junctions in epithelial cells (C. Royer et al., PLOS ONE, 2014). We have verified in MEFs derived from *ASPP2*^{RAKA/RAKA} embryos that the interaction between *ASPP2*^{RAKA} and PP1 is abrogated as expected. We now also include a figure showing that *ASPP2*^{RAKA} proteins localise at the apical junctions in the VE as expected (Extended Data Fig. 4b and below). Together, this suggests that phenotypes observed in those mutants are likely due to the inability of ASPP2 to recruit PP1.

3) Quantitative value of mechanical stress

There is no biophysical evidence demonstrating that defected tissues and regions show stiffer mechanical stress than other unaffected regions in *Aspp2* mutant embryos. If their hypothesis was correct, cell cortex tension of *Aspp2* mutant tissues and cells would be softer or more fragile than those of the wild type. You should measure the stiffness of the affected tissues quantitatively with an atomic force microscope directly or with a tension sensor imaging system indirectly. Analysis with p-Myosin expression is not quantitative.

We thank the reviewer for this comment. Our initial conclusion that the epiblast was under more mechanical stress was derived from confinement experiments using *ASPP2*^{RAKA/RAKA} mutants in a BALB/c background. We agree that direct evidence that the epiblast was under higher mechanical stress was lacking.

One cannot use techniques such as AFM to analyse tensions in this context because the apical surface of the epiblast is internal and therefore inaccessible to the AFM probe without disrupting the structure of the embryo. Instead, to address this concern, we developed an approach to use the FLIPPER-TR probe in combination with fluorescence lifetime imaging (Colom et al., Nature Chemistry, 2018) in the early post-implantation embryo.

We found that lifetimes were highest at the apical surface of the epiblast around the proamniotic cavity in comparison to the apical surface of the embryonic VE or extraembryonic VE at E6.5, suggesting that tensions are highest there (Fig. 5a,b; Extended Data Fig. 5 and below). This supports our conclusion that *ASPP2* is required in tissues that are under higher mechanical stress.

We also found that in *ASPP2*^{Epi Δ E4/ Δ E4} embryos, the epiblasts exhibited overall lower lifetimes, consistent with the prediction of this reviewer that the epiblast “would be softer or more fragile than those of the wild type.” Our data suggest that in the absence of *ASPP2* and organised apical F-actin, epiblast cell membranes experience overall less tensions. In addition, in contrast to control embryos in which higher tension was uniformly distributed along the apical surface of the epiblast, *ASPP2*^{Epi Δ E4/ Δ E4} embryos did not exhibit a clearly organised pattern, suggesting that *ASPP2*, through its regulation of apical F-actin organisation, is able to localise tensions apically (Fig. 6i; Extended Data Fig. 6g,h,i and below).

4) Primary defects of *Aspp2* mutant embryos.

It remains unclear if mutant defects are primarily due to failure in epithelial polarity, cell growth/apoptosis, or mechanical stress. How epithelial polarity is linked to amniotic cavity formation? They should clarify whether the phenotypes of *Aspp2* mutant embryos in Figure 5 are caused by the overgrowth/apoptosis or mechanical failure.

As described in the response to Reviewer 1, we have now performed additional experiments to test whether any of the differences seen in *ASPP2* mutants could be due to excess proliferation or reduced apoptosis and do not find support for either possibility.

We performed phospho-histone H3 and cleaved Caspase-3 Immunostainings in control and *ASPP2*^{EpiΔE4/ΔE4} embryos. At E6.5, we could not find significant differences in the proliferation rate of epiblast cells between control and *ASPP2*^{EpiΔE4/ΔE4} embryos (Extended Data Fig. 2f and below). We therefore rule out the possibility that cells escaped into the proamniotic cavity because of an excess of proliferation in the epiblast.

Additionally, very few epiblast cells were cleaved Caspase-3-positive in either control or *ASPP2*^{EpiΔE4/ΔE4} embryos at E6.5 (Extended Data Fig. 2h and below). We therefore conclude that it is extremely unlikely

those changes in the proportion of apoptotic cells are at the origin of the epiblast architecture defects we observe in *ASPP2* mutant embryos.

Besides, the authors might misinterpret normal epithelial-mesenchymal transition (EMT) and overgrowth of mesodermal cells as defects in epithelial polarity (in Figure 4). To clarify two phenomena of overgrowth of mesodermal cells and polarity failure explicitly, they could analyze specific markers expression such as EMT- and germ layer-specific markers including E-cadherin, N-cadherin, Tbr1, and so on.

As suggested, we have looked at several germ layer specific markers in the *ASPP2*^{RAKA/RAKA} mutants in a BALB/c background, including GATA6 (Visceral endoderm, Fig. 5d,e), AMOT (AVE, Fig. 4a), T (mesoderm, Fig. 4b), SOX2 (Ectoderm, Fig. 7j) and all are expressed normally, suggesting that *ASPP2* is not required for germ layer specification. In addition, our preliminary scRNAseq data suggests that the expression levels of EMT-related mRNA are unchanged between wild type and *ASPP2*^{RAKA/RAKA} cells, as detailed in the response above to Reviewer 2.

5) Phenotypes of mutant visceral endoderm

Except for epiblast defects, extraembryonic specific markers should be examined for mutant embryos, in particular, for the visceral endoderm (VE) layer; AVE/ DVE, pan-VE markers, and so on.

We now provide additional marker analyses showing that extraembryonic tissues are properly specified in mutants:

- AMOT showing normal AVE migration in *ASPP2*^{RAKA/RAKA} embryos (Fig 4a).
- GATA6 showing correct specification of the VE layer (Extended Data fig. 2c and below)
- SOX17 showing specification of the definitive endoderm (Extended Data Fig. 2g and below).

Additionally, our preliminary scRNAseq dataset shows very little transcriptional changes between cells from wild type and $ASPP2^{RAKA/RAKA}$ E7.5 embryos. Our evidence suggests that these extraembryonic tissues are specified normally in ASPP2 mutant embryos.

Additionally, since *Aspp2* is expressed in the VE layer (in Figure 6), epithelial polarity defects can be observed in the mutant VE cells. Moreover, they could identify the epithelial polarity defects more specifically than epiblast cells because mutant VE cells appear to show little overgrowth as compared to epiblast cells do. If you could not find polarity defects in the *Aspp2* mutant VE layer, *Aspp1* may complement the function of the *Aspp2* gene in VE cells.

Since we did not report any defects in the VE in our original submission, we interpret the reviewer's comments above as follows. When they say "epithelial polarity defects can be observed in the mutant VE cells" they are suggesting that defects 'might be expected in the VE'. Similarly, when they say 'Moreover, they could identify the epithelial polarity defects', they mean 'we ought to be able to identify polarity defects'. In response to this comment, we have now looked in more detail at the VE of ASPP2 mutant embryos and do not detect any polarity defects. To more clearly demonstrate the absence of polarity defects, we now show a 3D opacity rendering of Par6 and F-actin localisation on the outer surface of wild type and $ASPP2^{\Delta E4/\Delta E4}$ embryos (Extended Data Fig. 2c).

As mentioned above in reply to this reviewer's first question, there is no rationale to explain how ASPP1 could compensate for the absence of ASPP2 in the VE. Furthermore, compensation of ASPP1 need not be the only mechanism whereby the VE might be 'protected' from the lack of ASPP2 function. As described above and now demonstrated by our tension data, the epiblast is under considerably greater tension than the VE, making it potentially more susceptible to a loss of ASPP2. Furthermore, differences in expression of other molecules that potentially work with ASPP2 (such as SHROOM), could also account for difference in requirement for ASPP2 function in different embryonic tissues. We have now discussed these possibilities in greater detail in the revised manuscript.

Reviewers' Comments:

Reviewer #1:

Remarks to the Author:

The authors have satisfactorily responded to my review critiques.

Reviewer #2:

Remarks to the Author:

As I noted in my original review, maintaining epithelial architecture and cell polarity are critical for normal embryonic development and homeostasis and are often disrupted during oncogenesis. Here the authors present exciting new data analyzing the role of the protein ASPP2 in early mouse development, revealing striking roles in cell reintegration after cell division and in maintaining epithelial architecture. They combine stunning imaging of mouse embryos with careful quantitative analysis to dig deeply into the cell biological basis of the embryonic phenotypes. The authors have addressed all of my original concerns and I think this revised manuscript will be of broad interest to the cell and developmental biology community.

Reviewer #3:

Remarks to the Author:

The authors have clarified many of points I requested at the first review. Specifically, FLIM imaging data are quite convincing. Now, I think the revised manuscript is greatly improved. I will support the publication.

RESPONSE REVIEWER COMMENTS

Reviewer #1 (Remarks to the Author):

The authors have satisfactorily responded to my review critiques.

Reviewer #2 (Remarks to the Author):

As I noted in my original review, maintaining epithelial architecture and cell polarity are critical for normal embryonic development and homeostasis and are often disrupted during oncogenesis. Here the authors present exciting new data analyzing the role of the protein ASPP2 in early mouse development, revealing striking roles in cell reintegration after cell division and in maintaining epithelial architecture. They combine stunning imaging of mouse embryos with careful quantitative analysis to dig deeply into the cell biological basis of the embryonic phenotypes. The authors have addressed all of my original concerns and I think this revised manuscript will be of broad interest to the cell and developmental biology community.

Reviewer #3 (Remarks to the Author):

The authors have clarified many of points I requested at the first review. Specifically, FLIM imaging data are quite convincing. Now, I think the revised manuscript is greatly improved. I will support the publication.

We thank the reviewers for their constructive criticism which have helped us improve our manuscript.